# Estimating individual treatment effects under unobserved confounding using binary instruments

## Abstract

Estimating individual treatment effects (ITEs) from observational data is relevant in many fields such as personalized medicine. However, in practice, the treatment assignment is usually confounded by unobserved variables and thus introduces bias. A remedy to remove the bias is the use of instrumental variables (IVs). Such settings are widespread in medicine (e.g., trials where compliance is used as binary IV). In this paper, we propose a novel, multiply robust machine learning framework, called MRIV, for estimating ITEs using binary IVs and thus yield an unbiased ITE estimator. Different from previous work for binary IVs, our framework estimates the ITE directly via a pseudo outcome regression. (1) We provide a theoretical analysis where we show that our framework yields multiply robust convergence rates: our ITE estimator achieves fast convergence even if several nuisance estimators converge slowly. (2) We further show that our framework asymptotically outperforms state-of-the-art plug-in IV methods for ITE estimation. (3) We build upon our theoretical results and propose a tailored deep neural network architecture called MRIV-Net for ITE estimation using binary IVs. Across various computational experiments, we demonstrate empirically that our MRIV-Net achieves state-of-the-art performance. To the best of our knowledge, our MRIV is the first multiply robust machine learning framework tailored to estimating ITEs in the binary IV setting.

## 1 Introduction

Individual treatment effects (ITEs) are relevant across many disciplines such as marketing [41] and personalized medicine [51]. Knowledge about ITEs provides insights into the heterogeneity of treatment effects, and thus help in potentially better treatment decisions.

Many recent works that use machine learning to estimate ITEs are based on the assumption of unconfoundedness [1, 15, 27, 36, 42], In practice, however, this assumption is often violated because it is common that some confounders are not reported in the data. Typical examples are race, income, gender, or the socioeconomic status of patients, which are not stored in medical files. If the confounding is sufficiently strong, standard methods for estimating ITEs suffer from confounding bias [31], which may lead to inferior treatment decisions.

To handle unobserved confounders, instrumental variables (IVs) can be leveraged to relax the assumption of unconfoundedness and still compute reliable ITE estimates. IV methods were originally developed in economics [48], but, only recently, there is a growing interest in combining IV methods with machine learning (see Sec. 3). Importantly, IV methods outperform classical ITE estimators if a sufficient amount of confounding is not observed [17]. We thus aim at estimating ITEs from observational data under unobserved confounding using IVs.

In this paper, we consider the setting where a single binary instrument is available. This setting is widespread in personalized medicine (and other applications such as marketing or public policy) [9]. In fact, the setting is encountered in essentially all observational or randomized studies with observed non-compliance [19]. As an example, consider a randomized controlled trial (RCT), where treatments are randomly assigned to patients and their outcomes are observed. Due to some potentially unobserved confounders (e.g., income, education), some patients refuse to take the treatment initially assigned to them. Here, the treatment assignment serves as a binary IV. Moreover, such RCTs have been widely used by public decision-makers, e.g., to analyze the effect of health insurance on health outcome (see the so-called *Oregon health insurance experiment*) [16] or the effect of military service on lifetime earnings [2].

We propose a novel machine learning framework (called MRIV) for estimating ITEs using binary IVs. Our framework takes an initial ITE estimator and nuisance parameter estimators as input to perform a pseudo-outcome regression. Importantly, our framework uses a multiply robust parametrization of the efficient influence function as pseudo outcome.

We provide a theoretical analysis, where we use tools from [22] to show that our framework achieves a multiply robust convergence rate, i.e., our MRIV converges with a fast rate even if several nuisance parameters converge slowly. We further show that, compared to existing plug-in IV methods, the performance of our framework is asymptotically superior. Finally, we leverage our framework and, on top of it, build a tailored deep neural network called MRIV-Net.

**Differences to existing literature**: Our framework is **multiply robust**[1], i.e., it is consistent in the union of three different model specifications. This is different from existing methods for ITE estimation using IVs, which are only **doubly robust** (e.g., Syrgkanis et al. [40]) or plug-in estimators [5, 19].

**Contributions:**[2] (1) We propose a novel, multiply robust machine learning framework (called MRIV) to learn the ITE using the binary IV setting. To the best of our knowledge, ours is the first that is multiply robust, i.e., consistent in the union of three model specifications. For comparison, existing works for ITE estimation are only double robust [45, 40]. (2) We prove that MRIV achieves a multiply robust convergence rate. This is different to methods for IV settings which are only doubly robust, such as [40]. We further show that our MRIV is asymptotically superior to existing plug-in estimators. (3) We propose a tailored deep neural network, called MRIV-Net, which builds upon our framework to estimate ITEs. We demonstrate that MRIV-Net achieves state-of-the-art performance.

## 2 Problem setup

**Data generating process:** We observe data $\mathcal{D} = (x_i, z_i, a_i, y_i)_{i=1}^n$ consisting of $n \in \mathbb{N}$ observations of the tuple $(X, Z, A, Y)$. Here, $X \in \mathcal{X}$ are observed confounders, $Z \in \{0, 1\}$ is a binary instrument, $A \in \{0, 1\}$ is a binary treatment, and $Y \in \mathbb{R}$ is an outcome of interest. Furthermore, we assume the existence of unobserved confounders $U \in \mathcal{U}$, which affect both the treatment $A$ and the outcome $Y$. The causal graph is shown in Fig. 1.

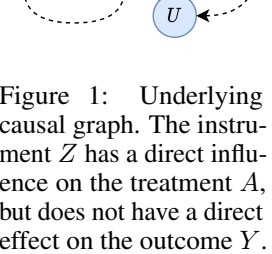

Figure 1: Underlying causal graph. The instrument $Z$ has a direct influence on the treatment $A$, but does not have a direct effect on the outcome $Y$. Note that we allow for unobserved confounders for both $Z$–$A$ (dashed line) and $A$–$Y$ (given by $U$). Our setting is general in that $U$ can be correlated or uncorrelated with the observed confounders X.

**Applicability:** Our proposed framework is widely applicable in practice, namely to all settings with the above data generating process. This includes both (1) observational data and (2) RCTs with non-compliance. For (1), observational data is commonly encountered in, e.g., personalized medicine. Here, modeling treatments as binary variables is consistent with previous literature on causal effect estimation and standard in medical practice [33]. For (2), our setting is further encountered in RCTs when the instrument $Z$ is a randomized treatment assignment but individuals do not comply with their treatment assignment. Such RCTs have been extensively used by public decision-makers, e.g.,

---

[1] For a detailed introduction to multiple robustness and its importance in treatment effect estimation, we refer to [46], Section 4.5.

[2] Codes are in the supplementary materials. Codes are also available at https://anonymous.4open.science/r/MRIV-Net-0AC4 (Upon acceptance, we replace the link and point to a public GitHub repository).

to analyze the effect of health insurance on health outcome [16] or the effect of military service on lifetime earnings [2].

We build upon the potential outcomes framework [34] for modeling causal effects. Let $Y(a, z)$ denote the potential outcome that would have been observed under $A = a$ and $Z = z$. Following previous literature on IV estimation [45], we impose the following standard IV assumptions on the data generating process.

**Assumption 1** (Standard IV assumptions [45, 47]). We assume: (1) *Exclusion:* $Y(a, z) = Y(a)$ for all $a, z \in \{0, 1\}$, i.e., the instrument has no direct effect on the patient outcome; (2) *Independence:* $Z \perp\!\!\!\perp U \mid X$; (3) *Relevance:* $Z \not\perp\!\!\!\perp A \mid X$, (iv) *The model includes all $A$–$Y$ confounder:* $Y(a) \perp\!\!\!\perp (A, Z) \mid (X, U)$ for all $a \in \{0, 1\}$.

Assumption 1 is standard for IV methods and fulfilled in practical settings where IV methods are applied [2, 4, 19]. Note that Assumption 1 does not prohibit the existence of unobserved $Z$–$A$ confounders. On the contrary, it merely prohibits the existence of unobserved counfounders that affect all $Z$, $A$, and $Y$ simultaneously, as it is standard in IV settings [47]. A practical and widespread example where Assumption 1 is satisfied are randomized controlled trials (RCTs) with non-compliance [19]. Here, the treatment assignment $Z$ is randomized, but the actual relationship between treatment $A$ and outcome $Y$ may still be confounded. For instance, in the *Oregon health insurance experiment* [16], people were given access to health insurance ($Z$) by a lottery with aim to study the effect of health insurance ($A$) on health outcome ($Y$) [16]. Here, non-compliance information is observed because the lottery winners needed to sign up for health insurance.

**Objective:** In this paper, we are interested in estimating the *individual treatment effect* (ITE)

$$\tau(x) = \mathbb{E}[Y(1) - Y(0) \mid X = x]. \tag{1}$$

If there is no unobserved confounding ($U = \emptyset$), the ITE is identifiable from observational data under mild positivity assumptions [36]. However, in practice, it is often unlikely that all confounders are observable. To account for this, we leverage the instrument $Z$ to identify the ITE. We state the following assumption for identifiability.

**Assumption 2** (Identifiability of the ITE [45]). At least one of the following two statements holds true: (1) $\mathbb{E}[A \mid Z = 1, X, U] - \mathbb{E}[A \mid Z = 0, X, U] = \mathbb{E}[A \mid Z = 1, X] - \mathbb{E}[A \mid Z = 0, X]$; or (2) $\mathbb{E}[Y(1) - Y(0) \mid X, U] = \mathbb{E}[Y(1) - Y(0) \mid X]$.

**Example:** Assumption 1 holds is when the function $f(a, X, U) = \mathbb{E}[Y(a) \mid X, U]$ is additive with respect to $a$ and $U$, e.g., $f(a, X, U) = g(a, X) + h(U)$ for measurable functions $h$ and $g$.

Under Assumptions 1 and 2, the ITE is identifiable [45]. It can be written as

$$\tau(x) = \frac{\mu_1^Y(x) - \mu_0^Y(x)}{\mu_1^A(x) - \mu_0^A(x)} = \frac{\delta_Y(x)}{\delta_A(x)}, \tag{2}$$

where $\mu_i^Y(x) = \mathbb{E}[Y \mid Z = i, X = x]$ and $\mu_i^A(x) = \mathbb{E}[A \mid Z = i, X = x]$. Even if Assumption 2 does not hold, the quantity on the right-hand side of Eq. (2) still allows for interpretation. If no unobserved $Z$–$A$ confounders exist, it can be interpreted as conditional version of the *local average treatment effect* (LATE) [19, 5] under a monotonicity assumption. Furthermore, under a no-current-treatment-value-interaction assumption, it can be interpreted as conditional *treatment effect on the treated* (ETT) [45]. [3] This has an important implication for our results: If Assumption 2 does not hold in practice, our estimates still provide conditional LATE or ETT estimates under the respective assumptions because they are based on Eq. (2). If Assumption 2 does hold, all three – i.e., ITE, conditional LATE, and ETT – coincide [45].

## 3 Related work

**ITE methods without unconfoundedness:** Various machine learning methods for estimating ITEs *without* unobserved confounding have been proposed in recent literature [1, 15, 25, 27, 36, 42, 52,

---

[3]The conditional LATE measures the ITE for individuals which are part of the complier subpopulation, i.e., the subpopulation for whom $A(Z = 1) > A(Z = 0)$. The conditional ETT measures the ITE for treated individuals.

53]. To remove plug-in bias, the DR-learner performs a second stage regression on the uncentered influence function of the average treatment effect [22, 14]. However, under unobserved confounding, all of these methods are biased (see Appendix). As a result, this hampers their performance in our setting.

**ITE methods for unobserved confounding:** There is a rich literature for causal effect estimation under unobserved confounding. Methods include deconfounding methods [46, 7, 18], proxy learning methods [13, 49], causal sensitivity analysis [21, 20], and IV methods. IV methods address the problem of unobserved confounding by exploiting the variance in treatment and outcome induced by the instruments. Traditionally, two-stage least squares (2SLS) has been used for estimating causal effects [48, 4]. 2SLS was originally developed in economics, and follows a two-stage procedure: it performs a first stage regression of treatment $A$ on the instrument $Z$, and then uses the fitted values for a second stage regression to predict the outcome $Y$. Several nonparametric methods have been developed in econometric to generalize 2SLS in order to account for non-linearities within the data [28, 44], yet these are limited to low-dimensional settings.

Only recently, machine learning has been integrated into IV methods. These are: [37] and [50] generalize 2SLS by learning complex feature maps using kernel methods and deep learning, respectively. [17] adopts a two-stage neural network architecture that performs the first stage via conditional density estimation. [6] and [40] leverage moment conditions for IV estimation. However, the aforementioned methods are not specifically designed for the binary IV setting but, rather, for multiple IVs or treatment scenarios. In particular, they impose stronger assumptions such as additive confounding in order to identify the ITE. Note that additive confounding is a special case of when our Assumption 2 holds. Moreover, they are not multiply robust: Even though doubly robust IV methods have been proposed (e.g., Syrgkanis et al. [40]), these methods are not consistent in the union of more than two model specifications [45]. We provide more details below.

**Doubly robust IV methods:** Doubly robust estimators are commonly used in causal inference as they allow for consistent estimation under model misspecification and fast convergence rates [22]. Recently, they also have been adopted for IV settings: [23] proposes a pseudo regression estimator for the local average treatment effect using continuous instruments, which has been extended to individual effects by [35]. Furthermore, [38] uses a doubly robust approach to estimate average compiler parameters. Finally, Ogburn et al. [29] and Syrgkanis et al. [40] propose doubly robust ITE estimators in the IV setting which both rely on doubly robust parametrizations of the uncentered efficient influence function [30]. However, these estimators are not multiply robust in the sense that they are consistent in the union of more than two model specifications [45].

Table 1: Key methods for causal effect estimation with IVs. This paper: Multiply robustness for ITEs.

| Robustness | Estimand | ATE | ITE |
|---|---|---|---|
| Doubly robust | | Okui et al. [30] | Syrgkanis et al. [40] |
| Multiply robust | | Wang et al. [45] | **MRIV** (ours) |

**Multiply robust IV methods:** Multiply robust estimators for IV settings have been proposed only for average treatment effects (ATEs) [45] and optimal treatment regimes [12] but not for ITEs. In particular, Wang et al. [45] derive a multiply robust parametrization of the efficient influence function for the ATE. However, there exists no similar approach for ITE estimation (see Table 1).

We provide a detailed, technical comparison of existing methods and our framework in Appendix E.

**Binary IVs:** In the binary IV setting, current methods proceed by estimating $\mu_i^Y(x)$ and $\mu_i^A(x)$ separately, before plugging them in Eq. 2 [19, 3, 5]. As result, these suffer from plug-in bias and do *not* offer robustness properties.

**Research gap:** To the best of our knowledge, there exists no method for ITE estimation under unobserved confounding that is *multiply robust*. To fill this gap, we propose MRIV: a *multiply robust* machine learning framework tailored to the binary IV setting. For this, we build upon the approach by Kennedy [22] to derive robust convergence rates, yet this approach has not been adapted to IV settings, which is our contribution.

# 4 MRIV for estimating ITEs using binary instruments

In the following, we present our MRIV framework for estimating ITEs under unobserved confounding (Sec. 4.1). We then derive an asymptotic convergence rate for MRIV (Sec. 4.2) and finally use our framework to develop a tailored deep neural network called MRIV-Net (Sec. 4.4).

## 4.1 Framework

**Motivation:** A naïve approach to estimate the ITE is to leverage the identification result in Eq. (2). Assuming that we have estimated the nuisance components $\hat{\mu}_i^Y$ and $\hat{\mu}_i^A$ for $i \in \{0, 1\}$, we can simply plug them into Eq. (2) to obtain the so-called (plug-in) Wald estimator $\hat{\tau}_W(x)$ [43].

However, in practice, the true ITE curve $\tau(x)$ is often simpler (e.g., smoother, more sparse) than its complements $\mu_i^Y(x)$ or $\mu_i^A(x)$ [25]. In this case, $\hat{\tau}_W(x)$ is inefficient because it models all components separately, and, to address this, our proposed framework estimates $\tau$ directly using a pseudo outcome regression.

**Overview:** We now propose MRIV. MRIV is a two-stage meta learner that takes any base method for ITE estimation as input. For instance, the base ssssmethod could be the Wald estimator from Eq. (2), any other IV method such as 2SLS, or a deep neural network (as we propose in our MRIV-Net later in Sec. 4.4). In Stage 1, MRIV produces nuisance estimators $\hat{\mu}_0^Y(x)$, $\hat{\mu}_0^A(x)$, $\hat{\delta}_A(x)$, and $\hat{\pi}(x)$, where $\hat{\pi}(x)$ is an estimator of the propensity score $\pi(x) = \mathbb{P}(Z = 1 \mid X = x)$. In Stage 2, MRIV estimates $\tau(x)$ directly using a pseudo outcome $\hat{Y}_{\mathrm{MR}}$ as a regression target.

Given an arbitrary initial ITE estimator $\hat{\tau}_{\mathrm{init}}(x)$ and nuisance estimates $\hat{\mu}_0^Y(x)$, $\hat{\mu}_0^A(x)$, $\hat{\delta}_A(x)$, and $\hat{\pi}(x)$, we define the pseudo outcome

$$\hat{Y}_{\mathrm{MR}} = \left( \frac{Z - (1 - Z)}{\hat{\delta}_A(X)} \right) \left( \frac{Y - \left( \hat{\mu}_0^Y(X) + \hat{\tau}_{\mathrm{init}}(X) \left( A - \hat{\mu}_0^A(X) \right) \right)}{Z\,\hat{\pi}(X) + (1 - Z)(1 - \hat{\pi}(X))} \right) + \hat{\tau}_{\mathrm{init}}(X). \tag{3}$$

The pseudo outcome $\hat{Y}_{\mathrm{MR}}$ in Eq. (3) is a multiply robust parameterization of the (uncentered) efficient influence function for the average treatment effect $\mathbb{E}_X[\tau(X)]$ (see the derivation in [45]). The initial estimator $\hat{\tau}_{\mathrm{init}}(X)$ is corrected by a weighted difference of the observed outcome $Y$ and the term $\hat{\mu}_0^Y(X) + \hat{\tau}_{\mathrm{init}}(X) \left( A - \hat{\mu}_0^A(X) \right)$. Individuals $X$ with small $\hat{\delta}_A(X)$ (large estimated compliance) or small/large $\pi(X)$ (i.e., low/high probability of receiving treatment $Z$) receive a larger correction.

Once we have obtained the pseudo outcome $\hat{Y}_{\mathrm{MR}}$, we regress it on $X$ to obtain the Stage 2 MRIV estimator $\hat{\tau}_{\mathrm{MRIV}}(x)$ for $\tau(x)$. The pseudocode for MRIV is given in Algorithm 1. MRIV can be interpreted as a way to remove plug-in bias from $\hat{\tau}_{\mathrm{init}}(x)$ via the efficient influence function [14]

---

**Algorithm 1:** MRIV

**Input:** data $(X, Z, A, Y)$, initial ITE estimator $\hat{\tau}_{\mathrm{init}}(x)$
// Stage 1: Estimate nuisance components
$\hat{\pi}(x) \leftarrow \hat{\mathbb{E}}[Z \mid X = x], \quad \hat{\mu}_0^Y(x) \leftarrow \hat{\mathbb{E}}[Y \mid X = x, Z = 0], \quad \hat{\mu}_0^A(x) \leftarrow \hat{\mathbb{E}}[A \mid X = x, Z = 0]$
$\hat{\delta}_A(x) \leftarrow \hat{\mathbb{E}}[A \mid X = x, Z = 1] - \hat{\mathbb{E}}[A \mid X = x, Z = 0]$
// Stage 2: Pseudo outcome regression
$\hat{Y}_{\mathrm{MR}} \leftarrow \left( \frac{Z - (1 - Z)}{\hat{\delta}_A(X)} \right) \left( \frac{Y - A\,\hat{\tau}_{\mathrm{init}}(X) - \hat{\mu}_0^Y(X) + \hat{\mu}_0^A(X)\,\hat{\tau}_{\mathrm{init}}(X)}{Z\,\hat{\pi}(X) + (1 - Z)(1 - \hat{\pi}(X))} \right) + \hat{\tau}_{\mathrm{init}}(X)$
$\hat{\tau}_{\mathrm{MRIV}}(x) \leftarrow \hat{\mathbb{E}}[\hat{Y}_{\mathrm{MR}} \mid X = x]$

---

Using the fact that $\hat{Y}_{\mathrm{MR}}$ is a multiply robust parametrization of the efficient influence function, we derive a multiply robustness property of $\hat{\tau}_{\mathrm{MRIV}}(x)$.

**Theorem 1** (multiply robustness property). *Let $\hat{\mu}_0^Y(x)$, $\hat{\mu}_0^A(x)$, $\hat{\delta}_A(x)$, $\hat{\pi}(x)$, and $\hat{\tau}_{\mathrm{init}}(x)$ denote estimators of $\mu_0^Y(x)$, $\mu_0^A(x)$, $\delta_A(x)$, $\pi(x)$, and $\tau(x)$, respectively. Then, for all $x \in \mathcal{X}$, it holds that $\mathbb{E}[\hat{Y}_{\mathrm{MR}} \mid X = x] = \tau(x)$, if least one of the following conditions is satisfied: (1) $\hat{\mu}_0^Y = \mu_0^Y$, $\hat{\mu}_0^A = \mu_0^A$, $\hat{\delta}_A = \delta_A$, and $\hat{\tau}_{\mathrm{init}} = \tau$; or (2) $\hat{\pi} = \pi$ and $\hat{\delta}_A = \delta_A$; or (3) $\hat{\pi} = \pi$ and $\hat{\tau}_{\mathrm{init}} = \tau$.*

Theorem 1 implies that $\hat{\tau}_{\mathrm{MRIV}}(x)$ is consistent for $\tau(x)$ if either condition (1), (2), or (3) holds. As a result, our MRIV framework is *multiply robust* in the sense that our estimator, $\hat{\tau}_{\mathrm{MRIV}}(x)$, is consistent in the union of three different model specifications. Importantly, this is different from *doubly robust* estimators which are only consistent in the union of two model specifications [45].

**Example:** We illustrate the robustness under model specification (2) in an example. Let $\hat{\mu}_0^Y(x) = \hat{\mu}_0^A(x) = \hat{\tau}_{\mathrm{init}}(x) = 0$ be misspecified and let $\hat{\pi} = \pi$ and $\hat{\delta}_A = \delta_A$ be correctly specified. It follows $\mathbb{E}[\hat{Y}_{\mathrm{MR}} \mid X = x] = \frac{1}{\delta_A(X)} \mathbb{E}\left[ \frac{ZY - (1 - Z)Y}{Z\pi(x) + (1 - Z)(1 - \pi(x))} \mid X = x \right] = \frac{\mu_1^Y(x) - \mu_0^Y(x)}{\delta_A(X)} = \tau(x)$. This justifies the pseudo-outcome regression in last step of MRIV.

221 Our MRIV is directly applicable to RCTs with non-compliance: Then, the treatment assignment is
222 randomized and the propensity score $\pi(x)$ is known. Our MRIV framework can be thus adopted
223 by plugging in the known $\pi(x)$ into the pseudo outcome in Eq. (3). Moreover, $\hat{\tau}_{\text{MRIV}}(x)$ is already
224 consistent if either $\hat{\tau}_{\text{init}}(\text{x})$ or $\hat{\delta}_A(x)$ are.

## 4.2 Theoretical analysis

226 In the following, we derive the asymptotic convergence rate of MRIV under smoothness assumptions.
227 For this, we define $s$-smooth functions as functions contained in the Hölder class $\mathcal{H}(s)$, associated
228 with Stone's minimax rate [39] of $n^{-2s/(2s+p)}$, where $p$ is the dimension of $\mathcal{X}$.

229 **Assumption 3** (Smoothness). We assume that (1) the nuisance components $\mu_i^Y(\cdot)$ are $\alpha$-smooth,
230 $\mu_i^A(\cdot)$ and $\delta_A(\cdot)$ are $\beta$-smooth, and $\pi(\cdot)$ is $\delta$-smooth; (2) all nuisance components are estimated with
231 their respective minimax rate of $n^{\frac{-2k}{2k+p}}$, where $k \in \{\alpha, \beta, \delta\}$; and (3) the oracle ITE $\tau(\cdot)$ is $\gamma$-smooth
232 and the initial ITE estimator $\hat{\tau}_{\text{init}}$ converges with rate $r_\tau(n)$.

233 Assumption 3 for smoothness provides us with a way to quantify the difficulty of the underlying
234 nonparametric regression problems. Similar assumptions have been imposed for asymptotic analysis
235 of previous ITE estimators in [22, 15]. They can be replaced with other assumptions such as
236 assumptions on the level of sparsity of the ITE components. We also provide an asymptotic analysis
237 under sparsity assumptions (see Appendix B).

238 We additionally impose the following boundedness assumptions on the the underlying data generating
239 process and estimators.

240 **Assumption 4** (Boundedness). We assume that there exist constants $C, \rho, \widetilde{\rho}, \epsilon, K > 0$ such that for
241 all $x \in \mathcal{X}$ it holds that: (1) $|\mu_i^Y(x)| \leq C$; (2) $|\delta_A(x)| = |\mu_1^A(x) - \mu_0^A(x)| \geq \rho$ and $|\hat{\delta}_A(x)| \geq \widetilde{\rho}$;
242 (3) $\epsilon \leq \hat{\pi}(x) \leq 1 - \epsilon$; and (4) $|\hat{\tau}_{\text{init}}(x)| \leq K$.

243 Assumptions 4.1, 4.3, and 4.4 are standard and in line with previous works on theoretical analyses
244 of ITE estimators [15, 22]. Assumption 4.2 ensures that both the oracle ITE and the estimator are
245 bounded. Violations of Assumption 4.2 may occur when working with so-called "weak" instruments,
246 which are IVs that are only weakly correlated with the treatment. Using IV methods with weak
247 instruments should generally be avoided [26]. However, in many applications such as RCTs with
248 non-compliance, weak instruments are unlikely to occur as patients' decisions to follow the treatment
249 are generally correlated with the initial treatment assignments.

250 We state now our main theoretical result: an upper bound on the oracle risk of the MRIV estimator.
251 To derive our bound, we leverage the sample splitting approach from [22]. The approach in [22] has
252 been initially used to analyze the DR-learner for ITE estimation under unconfoundedness and allows
253 for the derivation of robust convergence rates. It has later been adapted to several other meta learners
254 [15], yet not for IV methods.

255 **Theorem 2** (Oracle upper bound under sample splitting). *Let $\mathcal{D}_\ell$ for $\ell \in \{1, 2, 3\}$ be independent*
256 *samples of size $n$. Let $\hat{\tau}_{init}(x)$, $\hat{\mu}_0^Y(x)$, and $\hat{\mu}_0^A(x)$ be trained on $\mathcal{D}_1$, and let $\hat{\delta}_A(x)$ and $\hat{\pi}(x)$ be*
257 *trained on $\mathcal{D}_2$. We denote $\hat{Y}_{\text{MR}}$ as the pseudo outcome from Eq. (3) and $Y_0$ as the corresponding*
258 *oracle. Let $\hat{\tau}_{\text{MRIV}}(x) = \hat{\mathbb{E}}_n[\hat{Y}_{\text{MR}} \mid X = x]$ and $\widetilde{\tau}_{\text{MRIV}}(x) = \hat{\mathbb{E}}_n[Y_0 \mid X = x]$ denote the (oracle)*
259 *pseudo outcome regression on $\mathcal{D}_3$ for some generic estimator $\hat{\mathbb{E}}_n[\cdot \mid X = x]$ of $\mathbb{E}[\cdot \mid X = x]$.*

260 *We assume that the second-stage estimator $\hat{\mathbb{E}}_n$ yields the minimax rate $n^{-\frac{2\gamma}{2\gamma+p}}$ and satisfies the fol-*
261 *lowing two assumptions from Kennedy [22]: (1) $\hat{\mathbb{E}}_n[W + c \mid X = x] = \hat{\mathbb{E}}_n[W \mid X = x] + c$*
262 *for any random $W$ and constant $c$ and (2) if $\mathbb{E}[W \mid X = x] = E[V \mid X = x]$, then*
263 $\mathbb{E}\left[\left(\hat{\mathbb{E}}_n[W \mid X = x] - \mathbb{E}[W \mid X = x]\right)^2\right] \asymp \mathbb{E}\left[\left(\hat{\mathbb{E}}_n[V \mid X = x] - \mathbb{E}[V \mid X = x]\right)^2\right]$. *Then, the*
264 *oracle risk is upper bounded by*

$$\mathbb{E}\left[(\hat{\tau}_{\text{MRIV}}(x) - \tau(x))^2\right] \lesssim n^{\frac{-2\gamma}{2\gamma+p}} + r_\tau(n)\left(n^{\frac{-2\beta}{2\beta+p}} + n^{\frac{-2\delta}{2\delta+p}}\right) + n^{-2\left(\frac{\alpha}{2\alpha+p} + \frac{\delta}{2\delta+p}\right)} + n^{-2\left(\frac{\beta}{2\beta+p} + \frac{\delta}{2\delta+p}\right)}.$$

265 *Proof.* See Appendix A. □

Recall that the first summand of the lower bound in Eq. (2) is the minimax rate for the oracle ITE $\tau(x)$ which cannot be improved upon. Hence, for a fast convergence rate of $\hat{\tau}_{\mathrm{MRIV}}(x)$, it is sufficient if either: (1) $r_\tau(n)$ decreases fast and $\delta$ is large; (2) $r_\tau(n)$ decreases fast and $\alpha$ and $\beta$ are large; or (3) all $\alpha$, $\beta$, and $\delta$ are large. This is in line with the multiply robustness property of MRIV and means that MRIV achieves a fast rate of convergence even if the initial estimator or several nuisance estimators converge slowly.

From the bound in Eq. (2), it follows that $\hat{\tau}_{\mathrm{MRIV}}(x)$ improves on the convergence rate of the initial ITE estimator $\hat{\tau}_{\mathrm{init}}(x)$ if its rate $r_\tau(n)$ is lower bounded by

$$r_\tau(n) \gtrsim n^{\frac{-2\gamma}{2\gamma+p}} + n^{-2\left(\frac{\alpha}{2\alpha+p}+\frac{\delta}{2\delta+p}\right)} + n^{-2\left(\frac{\beta}{2\beta+p}+\frac{\delta}{2\delta+p}\right)}. \tag{4}$$

Hence, our MRIV estimator is more likely to improve on the initial estimator for large $\alpha$, $\beta$, and $\delta$, i.e., if the nuisance components are smooth. Note that it is sufficient if either (1) *only* the propensity score $\pi(x)$ is relatively smooth (large $\delta$) *or* (2) that *all* other nuisance components are (large $\alpha$ *and* $\beta$). In fact, this is widely fulfilled in practice. For example, the former is fulfilled for RCTs with non-compliance, where $\pi(x)$ is often some known, fixed number $p \in (0,1)$. Hence, for RCTs with non-compliance, MRIV should (at least asymptotically) improve the performance of most estimators.

## 4.3 MRIV vs. Wald estimator

In the following, we compare $\hat{\tau}_{\mathrm{MRIV}}(x)$ to the Wald estimator $\hat{\tau}_{\mathrm{W}}(x)$. First, we derive corresponding upper bound under smoothness.

**Theorem 3** (Wald oracle upper bound). *Given estimators $\hat{\mu}_i^Y(x)$ and $\hat{\mu}_i^A(x)$. Let $\hat{\delta}_A(x) = \hat{\mu}_1^A(x) - \hat{\mu}_0^A(x)$ satisfy Assumption 4. Then, the oracle risk of the Wald estimator $\hat{\tau}_W(x)$ is bounded by*

$$\mathbb{E}\left[(\hat{\tau}_{\mathrm{W}}(x) - \tau(x))^2\right] \lesssim n^{-\frac{2\alpha}{2\alpha+p}} + n^{-\frac{2\beta}{2\beta+p}}. \tag{5}$$

*Proof.* See Appendix A. □

We now consider the MRIV estimator $\hat{\tau}_{\mathrm{MRIV}}(x)$ with $\hat{\tau}_{\mathrm{init}} = \hat{\tau}_{\mathrm{W}}(x)$, i.e., initialized with the Wald estimator (under sample splitting). Plugging the Wald rate from Eq. (5) into the Eq. (2) yields

$$\mathbb{E}\left[(\hat{\tau}_{\mathrm{MRIV}}(x) - \tau(x))^2\right] \lesssim n^{\frac{-2\gamma}{2\gamma+p}} + n^{\frac{-4\beta}{2\beta+p}} + n^{-2\left(\frac{\alpha}{2\alpha+p}+\frac{\beta}{2\beta+p}\right)} + n^{-2\left(\frac{\delta}{2\delta+p}+\frac{\alpha}{2\alpha+p}\right)} + n^{-2\left(\frac{\delta}{2\delta+p}+\frac{\beta}{2\beta+p}\right)}. \tag{6}$$

For $\alpha = \beta = \delta$, the rates of $\hat{\tau}_{\mathrm{MRIV}}(x)$ and $\hat{\tau}_{\mathrm{W}}(x)$ reduce to

$$\mathbb{E}\left[(\hat{\tau}_{\mathrm{MRIV}}(x) - \tau(x))^2\right] \lesssim n^{\frac{-2\gamma}{2\gamma+p}} + n^{\frac{-4\alpha}{2\alpha+p}} \quad \text{and} \quad \mathbb{E}\left[(\hat{\tau}_{\mathrm{W}}(x) - \tau(x))^2\right] \lesssim n^{\frac{-2\alpha}{2\alpha+p}}. \tag{7}$$

Hence, $\hat{\tau}_{\mathrm{MRIV}}(x)$ outperforms $\hat{\tau}_{\mathrm{W}}(x)$ asymptotically for $\gamma > \alpha$, i.e., when the ITE $\tau(x)$ is smoother than its components, which is usually the case in practice [25]. For $\gamma = \alpha$, the rates of both estimators coincide. Hence, we should expect MRIV to improve on the Wald estimator in real-world settings with sufficiently large sample size.

## 4.4 MRIV-Net

Based on our MRIV framwork, we develop a tailored deep neural network called MRIV-Net for ITE estimation using IVs. Our MRIV-Net produces both an initial ITE estimator $\hat{\tau}_{\mathrm{init}}(x)$ and nuisance estimators $\hat{\mu}_0^Y(x)$, $\hat{\mu}_0^A(x)$, $\hat{\delta}_A(x)$, and $\hat{\pi}(x)$.

For MRIV-Net, we choose deep neural networks for the nuisance components due to their predictive power and their ability to learn complex shared representations for several nuisance components. Sharing representations between nuisance components has been exploited previously for ITE estimation, yet only under unconfoundedness [36, 15]. Building shared representations is more efficient in finite sample regimes than estimating all nuisance components separately as they usually share some common structure.

In MRIV-Net, not all nuisance components should share a representation. Recall that, in Theorem 2, we assumed that (1) $\hat{\tau}_{\mathrm{init}}(x)$, $\hat{\mu}_0^Y(x)$, and $\hat{\mu}_0^A(x)$; and (2) $\hat{\delta}_A(x)$ and $\hat{\pi}(x)$ are trained on two independent samples in order to derive the upper bound on the oracle risk. Hence, we propose to build two separate representations $\Phi_1$ and $\Phi_2$, so that (i) $\Phi_1$ is used to learn $\hat{\tau}_{\mathrm{init}}(x)$, $\hat{\mu}_0^Y(x)$, and $\hat{\mu}_0^A(x)$, and (ii) $\Phi_2$ is used to learn $\hat{\delta}_A(x)$ and $\hat{\pi}(x)$.

308 This ensures that the nuisance estimators (1) share minimal information
309 with nuisance estimators (2) even though they are estimated on the same
310 data. Intuitively, this should lead to a faster decay of the oracle upper
311 bound (cf. [15]).

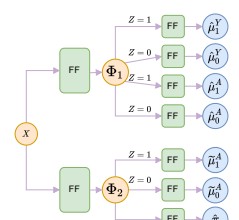

Figure 2: Architecture of MRIV-Net.

312 The architecture of MRIV-Net is shown in Fig. 2. MRIV-Net takes the
313 observed covariates $X$ as input to build the two representations $\Phi_1$ and
314 $\Phi_2$. The first representation $\Phi_1$ is used to output estimates $\hat{\mu}_1^Y(x)$, $\hat{\mu}_0^Y(x)$,
315 $\hat{\mu}_1^A(x)$, and $\hat{\mu}_0^A(x)$ of the ITE components. The second representation
316 $\Phi_2$ is used to output estimates $\widetilde{\mu}_1^A(x)$, $\widetilde{\mu}_0^A(x)$, and $\hat{\pi}(x)$. MRIV-Net is
317 trained by minimizing an overall loss

$$\mathcal{L}(\theta) = \sum_{i=1}^n \left[ \left( \hat{\mu}_{z_i}^Y(x_i) - y_i \right)^2 + \mathrm{BCE}\left( \hat{\mu}_{z_i}^A(x_i), a_i \right) + \mathrm{BCE}\left( \widetilde{\mu}_{z_i}^A(x_i), a_i \right) + \mathrm{BCE}\left( \hat{\pi}(x_i), z_i \right) \right], \quad (8)$$

318 where $\theta$ denotes the neural network parameters and BCE is the binary cross entropy loss. After
319 training MRIV-Net, we obtain the $\hat{\tau}_{\mathrm{init}}(x) = \frac{\hat{\mu}_1^Y(x) - \hat{\mu}_0^Y(x)}{\hat{\mu}_1^A(x) - \hat{\mu}_0^A(x)}$ and obtain the nuisance estimators $\hat{\mu}_0^Y(x)$,
320 $\hat{\mu}_0^A(x)$, $\hat{\delta}_A(x) = \widetilde{\mu}_1^A(x) - \widetilde{\mu}_0^A(x)$ and $\hat{\pi}(x)$. Then, we perform, we perform the pseudo regression
321 (Stage 2) of MRIV to obtain $\hat{\tau}_{\mathrm{MRIV}}(x)$.

322 **Implementation:** We use PyTorch Lightning for our implementation and train MRIV-Net with
323 the Adam optimizer [24]. Details on the network architecture and hyperparameter tuning are in
324 Appendix G. We perform both the training of MRIV-Net and the pseudo outcome regression on
325 the full training data. Needless to say, MRIV-Net can be easily adopted for sample splitting or
326 cross-fitting procedures as in [10], namely, by learning separate networks for each representation
327 $\Phi_1$ and $\Phi_2$. However, in our experiments, we do not use sample splitting or cross-fitting, as this can
328 affect the performance in finite sample regimes. Of note, our choice is consistent with previous work
329 [15].

## 5 Computational experiments

### 5.1 Simulated data

332 In causal inference literature, it is common practice to use simulated data for performance evaluations
333 [8, 15, 17]. Simulated data offers the crucial benefit that it provides ground-truth information on the
334 counterfactual outcomes and thus allows for direct benchmarking against the oracle ITE.

335 **Data generation:** We generate simulated data by sampling the oracle ITE $\tau(x)$ and the nuisance
336 components $\mu_i^Y(x)$, $\mu_i^A(x)$, and $\pi(x)$ from Gaussian process priors. Using Gaussian processes has
337 the following advantages: (1) It allows for a fair method comparison, as there is no need to explicitly
338 specify the nuisance components, which could lead to unwanted inductive biases favoring a specific
339 method; (2) the sampled nuisance components are non-linear and thus resemble real-world scenarios
340 where machine learning methods would be applied; and, (3) by sampling from the prior induced by
341 the Matérn kernel [32], we can control the smoothness of the nuisance components, which allows
342 us to confirm our theoretical results from Sec. 4.2. For a detailed description of our data generating
343 process, we refer to Appendix C.

344 **Baselines:** We compare our MRIV-Net with the following state-of-the-art baselines: (1) ITE methods
345 for unconfoundedness: **TARNet** [36] and TARNet combined with the **DR-learner** [22]; (2) general
346 IV methods: **2SLS** [48], kernel IV (**KIV**) [37], **DFIV** [50], **DeepIV** [17], **DeepGMM** [6], **DMLIV**
347 [40], and DMLIV combined with **DRIV** (as described in [40]); (3) the (plug-in) Wald estimator using
348 **linear models** and Bayesian additive regression trees (**BART**) [11]. Of note, the DR-learner assumes
349 unconfoundedness, which is why we only combine it TARNet in our experiments. Implementation
350 details regarding baselines and nuisance parameter estimation are in Appendix E. Note that many of
351 the baselines do not directly aim at ITE estimation but rather at counterfactual outcome prediction.
352 We nevertheless use these methods as baselines and, for this, obtain the ITE by taking the difference
353 between the predictions of the factual and counterfactual outcomes.

354 **Performance evaluation:** For all experiments, we use a 80/20 split as training/test set. We calcalute
355 the root mean squared errors (RMSE) between the ITE estimates and the oracle ITE on the test set.

We report the mean RMSE and the standard deviation over five data sets generated from random seeds.

**Results:** Table 2 shows the results for all baselines. Here, the DR-learner does not improve the performance of TAR-Net, which is reasonable as both the DR-learner and TARNet assume unconfoundedness and are thus biased in our setting. Our MRIV-Net outperforms all baselines. Our MRIV-Net also achieves a smaller standard deviation. For additional results, we refer to Appendix H.

We further compare the performance of two different meta-learner frameworks – DRIV [40] and our MRIV– across different base methods. The nuisance parameters are estimated using feed forward neural networks (DRIV) or TARNets with either binary or continuous outputs (MRIV). The results are in Table 3. Our MRIV improves over the variant without any meta-learner framework across all base methods (both in terms of RMSE and standard deviation).

Table 2: Performance comparison: our MRIV-Net vs. existing baselines.

| Method | $n = 3000$ | $n = 5000$ | $n = 8000$ |
|---|---|---|---|
| (1) STANDARD ITE | | | |
| TARNet [36] | $0.76 \pm 0.14$ | $0.70 \pm 0.12$ | $0.69 \pm 0.17$ |
| TARNet + DR [36, 22] | $0.78 \pm 0.10$ | $0.66 \pm 0.09$ | $0.70 \pm 0.10$ |
| (2) GENERAL IV | | | |
| 2SLS [47] | $1.22 \pm 0.23$ | $0.79 \pm 0.37$ | $1.12 \pm 0.29$ |
| KIV [37] | $1.54 \pm 0.53$ | $1.18 \pm 1.14$ | $3.80 \pm 4.71$ |
| DFIV [50] | $0.43 \pm 0.11$ | $0.40 \pm 0.21$ | $0.46 \pm 0.54$ |
| DeepIV [17] | $0.96 \pm 0.30$ | $0.28 \pm 0.09$ | $0.23 \pm 0.04$ |
| DeepGMM [6] | $0.95 \pm 0.38$ | $0.37 \pm 0.09$ | $0.42 \pm 0.14$ |
| DMLIV [40] | $1.92 \pm 0.71$ | $0.92 \pm 0.41$ | $1.14 \pm 0.24$ |
| DMLIV + DRIV [40] | $0.41 \pm 0.12$ | $0.22 \pm 0.04$ | $0.21 \pm 0.06$ |
| (3) WALD ESTIMATOR [43] | | | |
| Linear | $1.06 \pm 0.63$ | $0.62 \pm 0.22$ | $0.81 \pm 0.34$ |
| BART | $0.95 \pm 0.30$ | $0.63 \pm 0.33$ | $0.88 \pm 0.28$ |
| MRIV-Net (ours) | $\mathbf{0.26 \pm 0.11}$ | $\mathbf{0.15 \pm 0.03}$ | $\mathbf{0.13 \pm 0.03}$ |

Reported: RMSE for base methods (mean ± standard deviation). Lower = better (best in bold)

Table 3: Base model with different meta-learners (i.e., none, DRIV, and our MRIV).

| Meta-learners / Base methods | $n = 3000$ | | | $n = 5000$ | | | $n = 8000$ | | |
|---|---|---|---|---|---|---|---|---|---|
| | None | DRIV | MRIV (ours) | None | DRIV | MRIV (ours) | None | DRIV | MRIV (ours) |
| (1) STANDARD ITE | | | | | | | | | |
| TARNet [36] | $0.76 \pm 0.14$ | $\mathbf{0.31 \pm 0.05}$ | $0.34 \pm 0.13$ | $0.70 \pm 0.12$ | $\mathbf{0.17 \pm 0.06}$ | $\mathbf{0.17 \pm 0.05}$ | $0.69 \pm 0.17$ | $0.21 \pm 0.04$ | $\mathbf{0.16 \pm 0.04}$ |
| (2) GENERAL IV | | | | | | | | | |
| 2SLS [47] | $1.22 \pm 0.23$ | $0.40 \pm 0.11$ | $\mathbf{0.31 \pm 0.08}$ | $0.79 \pm 0.37$ | $0.17 \pm 0.09$ | $\mathbf{0.19 \pm 0.05}$ | $1.12 \pm 0.29$ | $0.21 \pm 0.05$ | $\mathbf{0.16 \pm 0.02}$ |
| KIV [37] | $1.54 \pm 0.53$ | $0.40 \pm 0.10$ | $\mathbf{0.39 \pm 0.11}$ | $1.18 \pm 1.14$ | $0.20 \pm 0.08$ | $\mathbf{0.17 \pm 0.06}$ | $3.80 \pm 4.71$ | $0.31 \pm 0.18$ | $\mathbf{0.28 \pm 0.19}$ |
| DFIV [50] | $0.43 \pm 0.11$ | $\mathbf{0.26 \pm 0.05}$ | $0.27 \pm 0.07$ | $0.40 \pm 0.21$ | $0.18 \pm 0.09$ | $\mathbf{0.16 \pm 0.04}$ | $0.46 \pm 0.54$ | $0.21 \pm 0.06$ | $\mathbf{0.18 \pm 0.05}$ |
| DeepIV [17] | $0.96 \pm 0.30$ | $0.27 \pm 0.03$ | $\mathbf{0.26 \pm 0.05}$ | $0.28 \pm 0.09$ | $0.18 \pm 0.08$ | $\mathbf{0.18 \pm 0.05}$ | $0.23 \pm 0.04$ | $0.21 \pm 0.03$ | $\mathbf{0.16 \pm 0.03}$ |
| DeepGMM [6] | $0.95 \pm 0.38$ | $0.40 \pm 0.15$ | $\mathbf{0.36 \pm 0.11}$ | $0.37 \pm 0.09$ | $0.24 \pm 0.12$ | $\mathbf{0.16 \pm 0.05}$ | $0.42 \pm 0.14$ | $0.21 \pm 0.03$ | $\mathbf{0.17 \pm 0.03}$ |
| DMLIV [40] | $1.92 \pm 0.71$ | $0.41 \pm 0.12$ | $\mathbf{0.37 \pm 0.11}$ | $0.92 \pm 0.41$ | $0.22 \pm 0.05$ | $\mathbf{0.16 \pm 0.05}$ | $1.14 \pm 0.24$ | $0.21 \pm 0.06$ | $\mathbf{0.18 \pm 0.05}$ |
| (3) WALD ESTIMATOR [43] | | | | | | | | | |
| Linear | $1.06 \pm 0.63$ | $0.42 \pm 0.15$ | $\mathbf{0.38 \pm 0.14}$ | $0.62 \pm 0.22$ | $0.19 \pm 0.09$ | $\mathbf{0.25 \pm 0.09}$ | $0.81 \pm 0.34$ | $0.19 \pm 0.06$ | $\mathbf{0.18 \pm 0.04}$ |
| BART | $0.95 \pm 0.30$ | $0.48 \pm 0.14$ | $\mathbf{0.46 \pm 0.12}$ | $0.63 \pm 0.33$ | $0.26 \pm 0.13$ | $\mathbf{0.20 \pm 0.07}$ | $0.88 \pm 0.28$ | $0.31 \pm 0.08$ | $\mathbf{0.29 \pm 0.04}$ |
| MRIV-Net\w network only (ours) | $0.39 \pm 0.13$ | $0.35 \pm 0.12$ | $\mathbf{0.26 \pm 0.11}$ | $0.31 \pm 0.04$ | $0.19 \pm 0.13$ | $\mathbf{0.15 \pm 0.03}$ | $0.26 \pm 0.06$ | $0.18 \pm 0.08$ | $\mathbf{0.13 \pm 0.03}$ |

Reported: RMSE (mean ± standard deviation). Lower = better (best improvement over none meta-learner in bold)

Furthermore, MRIV is clearly superior over DRIV. This demonstrates the effectiveness of our MRIV across different base methods (note: MRIV with an arbitrary base model is typically superior to DRIV with our custom network from above). MRIV-Net is overall best. We also performed additional experiments where we used cross-fitting approaches for both meta-learners (see Appendix I).

Table 4: Ablation study.

| Method | $n = 3000$ | $n = 5000$ | $n = 8000$ |
|---|---|---|---|
| MRIV-Net\w network only | $0.39 \pm 0.13$ | $0.31 \pm 0.04$ | $0.26 \pm 0.06$ |
| MRIV-Net\w single repr. | $0.28 \pm 0.12$ | $0.21 \pm 0.04$ | $0.32 \pm 0.10$ |
| MRIV-Net (ours) | $\mathbf{0.26 \pm 0.11}$ | $\mathbf{0.15 \pm 0.03}$ | $\mathbf{0.13 \pm 0.03}$ |

Reported: RMSE (mean ± standard deviation). Lower = better (best in bold)

**Ablation study:** Table 4 compares different variants of our MRIV-Net. These are: (1) MRIV but network only; (2) MRIV-Net with a single representation for all nuisance estimators; and (3) our MRIV-Net from above. We observe that MRIV-Net is best. This justifies our proposed network architecture for MRIV-Net. Hence, combing the result from above, our performance gain must be attributed to both our framework and the architecture of our deep neural network.

**Robustness checks for unobserved confounding and smoothness:** Here, we demonstrate the importance of handling unobserved confounding (as we do in our MRIV framework). For this, Fig. 3 plots the results for our MRIV-Net vs. standard ITE without customization for confounding (i.e., TARNet with and without the DR-learner) over over different levels of unobserved confounding. The RMSE of both TARNet variants increase almost linearly with

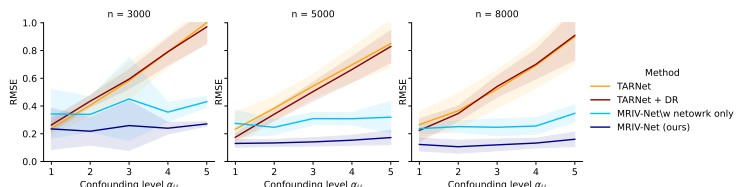

Figure 3: Results over different levels of confounding $\alpha_U$. Shaded area shows standard deviation.

increasing confounding. In contrast, the RMSE of our MRIV-Net only marginally. Even for low confounding regimes, our MRIV-Net performs competitively.

Fig. 4 varies the smoothness level. This is given by $\alpha$ of $\mu_i^Y(\cdot)$ (controlled by the Matérn kernel prior). Here, the performance decreases for the baselines, i.e., DeepIV and our network without MRIV framework. In contrast, the peformance of our MRIV-Net remains robust and outperforms the baselines. This confirms our theoretical results from above. It thus indicates that our MRIV framework works best when the oracle ITE $\tau(x)$ is smoother than.

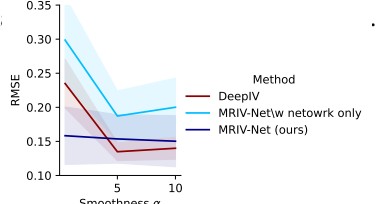

### 5.2 Case study with real-world data

**Setting:** We demonstrate effectiveness of our framework using a case study with real-world, medical data. Here, we use medical data from the so-called *Oregon health insurance experiment* (OHIE) [16]. It provides data for an RCT with non-compliance: In 2008, ∼30,000 low-income, uninsured adults in Oregon were offered participation in a health insurance program by a lottery. Individuals whose names were drawn could decide to sign up for health insurance. After a period of 12 months, in-person interviews took place to evaluate the health condition of the respective participant.

Figure 4: Results over different levels of smoothness $\alpha$ of $\mu_i^Y(\cdot)$, sample size $n = 8000$. Larger $\alpha$ = smoother. Shaded areas show standard deviation.

In our analysis, the lottery assignment is the instrument $Z$, the decision to sign up for health insurance is treatment $A$, and an overall health score is the outcome $Y$. We also include five covariates $X$ (age, gender, language, the number of emergency visits before the experiment, and the number of people the individual signed up with). It is important to include the latter in our analysis as it is the only variable influencing the propensity score. For details, we refer to Appendix D. We first estimate the ITE function and then report the treatment effect heterogeneity w.r.t. age and gender, while fixing the other covariates (i.e., we consider the English-speaking subpopulation with one emergency visit that signed up alone). We repeat the same procedure for our neural network architecture without the MRIV-Net framework and TARNet. The results are in Fig. 5.

**Results:** Our MRIV-Net estimates larger causal effects for an older age. In contrast, TARNet does not estimate positive ITEs even for an older age.

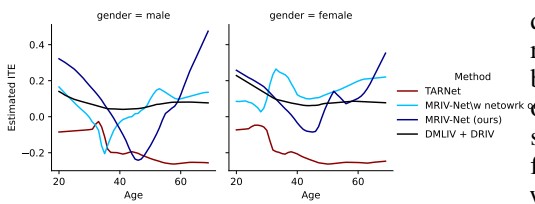

Figure 5: Results on real-world medical data.

Even though we cannot evaluate the estimation quality on real-world data, our estimates seem reasonable in light of the medical literature: the benefit of health insurance should increase with older age. This showcases that TARNet may suffer from bias induced by unobserved confounders. We also report the results for DRIV with DMLIV as base method, and observe that in contrast to MRIV-Net, the corresponding ITE does not vary much between ages. Interestingly, both our MRIV-Net estimate a somewhat smaller ITE for middle ages (around 30–50 yrs). One explanation might be that individual in this age group are more likely to have stable jobs and, thus, are also more likely to be able to afford medical care, decreasing the direct effect of health insurance on individuals health. In sum, the findings from our case study are of direct relevance for decision-makers in public health [19], and highlight the practical value of our framework.

## 6 Conclusion

In this paper, we propose MRIV-Net: a novel ITE estimator based on a deep neural network. Importantly, our estimator is consistent in the union of three models specifications and, therefore, *multiply robust*. This is a crucial difference to existing works: previously, existing ITE estimators (such es DRIV from Syrgkanis et al. [40]) were only *doubly robust*. We show both theoretically and empirically that MRIV-Net is state-of-the-art for estimating ITEs using binary IVs. For future work, it would be interesting to derive finite sample results for MRIV-Net, because our theoretical analysis is purely asymptotic. Furthermore, one could develop multiply robust estimators for other IV settings (e.g., multiple or continuous instruments and treatments).

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

# Estimating individual treatment effects under unobserved confounding using binary instruments
# Appendix

**Anonymous Author(s)**
Affiliation
Address
email

# Contents

 **A Proofs**

 We start by deriving an auxiliary Lemma. That is, we derive an explicit expression for the Stage 2
 oracle pseudo outcome regression $\mathbb{E}[\hat{Y}_0 \mid X = x]$ of MRIV.

**Lemma 4.**

$$
\mathbb{E}[\hat{Y}_0 \mid X = x]
$$

$$
=\frac{\pi(x)}{\hat{\delta}_A(x)\hat{\pi}(x)} \left(\mu_1^Y(x) - \mu_1^A(x)\,\hat{\tau}_{\text{init}}(x)\right) + \frac{(1 - \pi(x))}{\hat{\delta}_A(x)(1 - \hat{\pi}(x))} \left(\mu_0^A(x)\,\hat{\tau}_{\text{init}}(x) - \mu_0^Y(x)\right) \tag{1}
$$

$$
+ \frac{\hat{\mu}_0^A(x)\,\hat{\tau}_{\text{init}}(x) - \hat{\mu}_0^Y(x)}{\hat{\delta}_A(x)} \left(\frac{\pi(x)}{\hat{\pi}(x)} - \frac{1 - \pi(x)}{1 - \hat{\pi}(x)}\right) + \hat{\tau}_{\text{init}}(x)
$$

*Proof.*

$$
\mathbb{E}[\hat{Y}_0 \mid X = x] \tag{2}
$$

$$
=\pi(x)\mathbb{E}\left[\left.\frac{Y - A\,\hat{\tau}_{\text{init}}(X) - \hat{\mu}_0^Y(X) + \hat{\mu}_0^A(X)\,\hat{\tau}_{\text{init}}(X)}{\hat{\delta}_A(X)\,\hat{\pi}(X)} \right| X = x, Z = 1\right]
$$

$$
+ (1 - \pi(x))\mathbb{E}\left[\left.\frac{Y - A\,\hat{\tau}_{\text{init}}(X) - \hat{\mu}_0^Y(X) + \hat{\mu}_0^A(X)\,\hat{\tau}_{\text{init}}(X)}{\hat{\delta}_A(X)\,(1 - \hat{\pi}(X))} \right| X = x, Z = 0\right] + \hat{\tau}_{\text{init}}(x)
$$

$$
\tag{3}
$$

$$
=\frac{\pi(x)}{\hat{\delta}_A(x)\,\hat{\pi}(x)} \left(\mu_1^Y(x) - \mu_1^A(x)\,\hat{\tau}_{\text{init}}(x) - \hat{\mu}_0^Y(x) + \hat{\mu}_0^A(x)\,\hat{\tau}_{\text{init}}(x)\right)
$$

$$
+ \frac{1 - \pi(x)}{\hat{\delta}_A(x)\,(1 - \hat{\pi}(x))} \left(\mu_0^Y(x) - \mu_0^A(x)\,\hat{\tau}_{\text{init}}(x) - \hat{\mu}_0^Y(x) + \hat{\mu}_0^A(x)\,\hat{\tau}_{\text{init}}(x)\right) + \hat{\tau}_{\text{init}}(x) \tag{4}
$$

 Rearranging the terms yields the desired result. $\qquad\square$

 **A.1 Proof of Theorem 1 (multiple robustness property)**

 We use Lemma 4 to show that under each of the three conditions it follows that $\mathbb{E}[\hat{Y}_0 \mid X = x] = \tau(x)$.

1.

$$
\mathbb{E}[\hat{Y}_0 \mid X = x] \tag{5}
$$

$$
=\frac{\pi(x)}{\delta_A(x)\,\hat{\pi}(x)} \left(\mu_1^Y(x) - \mu_1^A(x)\,\tau(x) + \mu_0^A(x)\,\tau(x) - \mu_0^Y(x)\right)
$$

$$
+ \frac{(1 - \pi(x))}{\delta_A(x)\,(1 - \hat{\pi}(x))} \left(\mu_0^A(x)\,\tau(x) - \mu_0^Y(x) - \mu_0^A(x)\,\tau(x) + \mu_0^Y(x)\right) + \tau(x) \tag{6}
$$

$$
=\frac{\pi(x)}{\delta_A(x)\,\hat{\pi}(x)} \left(\delta_Y(x) - \delta_Y(x)\right) + \tau(x) = \tau(x). \tag{7}
$$

2.

$$
\mathbb{E}[\hat{Y}_0 \mid X = x] = \frac{\left(\mu_1^Y(x) - \mu_1^A(x)\,\hat{\tau}_{\text{init}}(x)\right)}{\delta_A(x)} + \frac{\left(\mu_0^A(x)\,\hat{\tau}_{\text{init}}(x) - \mu_0^Y(x)\right)}{\delta_A(x)} + \hat{\tau}_{\text{init}}(x)
$$

$$
\tag{8}
$$

$$
= \frac{\delta_Y(x) - \hat{\tau}_{\text{init}}(x)\,\delta_A(x)}{\delta_A(x)} + \hat{\tau}_{\text{init}}(x) = \tau(x). \tag{9}
$$

3.

$$
\mathbb{E}[\hat{Y}_0 \mid X = x] = \frac{\left(\mu_1^Y(x) - \mu_1^A(x)\,\tau(x)\right)}{\hat{\delta}_A(x)} + \frac{\left(\mu_0^A(x)\,\tau(x) - \mu_0^Y(x)\right)}{\hat{\delta}_A(x)} + \tau(x) \tag{10}
$$

$$
= \frac{\delta_Y(x)}{\hat{\delta}_A(x)} - \tau(x)\frac{\delta_A(x)}{\hat{\delta}_A(x)} + \tau(x) = \tau(x) \tag{11}
$$

 **A.2   Proof of Theorem 2 (Convergence rate of MRIV)**

24   To prove Theorem 2, we need an additional assumption on the second stage regression estimator $\hat{\mathbb{E}}_n$.
25   We refer to Kennedy [8] (Theorem 1) for a detailed discussion on this assumption.

26   **Assumption 5** (From Theorem 1 of Kennedy [8])**.**  The following two statements hold:

27       1. $\hat{\mathbb{E}}_n[W + c \mid X = x] = \hat{\mathbb{E}}_n[W \mid X = x] + c$ for any random $W$ and constant $c$

28       2. If $\mathbb{E}[W \mid X = x] = E[V \mid X = x]$ then

$$\mathbb{E}\left[\left(\hat{\mathbb{E}}_n[W \mid X = x] - \mathbb{E}[W \mid X = x]\right)^2\right] \asymp \mathbb{E}\left[\left(\hat{\mathbb{E}}_n[V \mid X = x] - \mathbb{E}[V \mid X = x]\right)^2\right].$$
(12)

29   *Proof of Theorem 2.*  Using Assumption 5, we can apply Theorem 1 of Kennedy [8] and obtain

$$\mathbb{E}\left[(\hat{\tau}_{\text{init}}(x) - \tau(x))^2\right] \lesssim \mathcal{R}(x) + \mathbb{E}\left[\hat{r}(x)^2\right],$$
(13)

30   where $\mathcal{R}(x) = \mathbb{E}\left[(\tilde{\tau}_{MR}(x) - \tau(x))^2\right]$ is the oracle risk of the second stage regression and $r(x) =$
31   $\mathbb{E}[\hat{Y}_0 \mid X = x] - \tau(x)$. We can apply Lemma 4 to obtain

$$
\begin{aligned}
\hat{r}(x) = {}& \frac{\pi(x)}{\hat{\delta}_A(x)\,\hat{\pi}(x)} \left(\mu_1^Y(x) - \mu_1^A(x)\,\hat{\tau}_{\text{init}}(x)\right) + \frac{(1 - \pi(x))}{\hat{\delta}_A(x)\,(1 - \hat{\pi}(x))} \left(\mu_0^A(x)\,\hat{\tau}_{\text{init}}(x) - \mu_0^Y(x)\right) \\
& + \frac{\hat{\mu}_0^A(x)\,\hat{\tau}_{\text{init}}(x) - \hat{\mu}_0^Y(x)}{\hat{\delta}_A(x)} \left(\frac{\pi(x)}{\hat{\pi}(x)} - \frac{1 - \pi(x)}{1 - \hat{\pi}(x)}\right) + \hat{\tau}_{\text{init}}(x) - \tau(x)
\end{aligned}
$$
(14)

$$
\begin{aligned}
= {}& \left(\frac{\mu_1^Y(x) - \mu_0^Y(x)}{\hat{\delta}_A(x)}\right) \frac{\pi(x)}{\hat{\pi}(x)} + \frac{\mu_0^Y(x) - \hat{\mu}_0^Y(x)}{\hat{\delta}_A(x)} \left(\frac{\pi(x)}{\hat{\pi}(x)} - \frac{1 - \pi(x)}{1 - \hat{\pi}(x)}\right) + (\hat{\tau}_{\text{init}}(x) - \tau(x)) \\
& + \left(\frac{(\mu_0^A(x) - \mu_1^A(x))\,\hat{\tau}_{\text{init}}(x)}{\hat{\delta}_A(x)}\right) \frac{\pi(x)}{\hat{\pi}(x)} + \frac{(\hat{\mu}_0^D(x) - \mu_0^D(x))\,\hat{\tau}_{\text{init}}(x)}{\hat{\delta}_A(x)} \left(\frac{\pi(x)}{\hat{\pi}(x)} - \frac{1 - \pi(x)}{1 - \hat{\pi}(x)}\right)
\end{aligned}
$$
(15)

$$
\begin{aligned}
= {}& \frac{\delta_Y(x)\,\pi(x)}{\hat{\delta}_A(x)\,\hat{\pi}(x)} + \frac{(\mu_0^Y(x) - \hat{\mu}_0^Y(x))\,(\pi(x) - \hat{\pi}(x))}{\hat{\delta}_A(x)\,\hat{\pi}(x)\,(1 - \hat{\pi}(x))} + (\hat{\tau}_{\text{init}}(x) - \tau(x)) \\
& - \frac{\delta_A(x)\,\pi(x)\,\hat{\tau}_{\text{init}}(x)}{\hat{\delta}_A(x)\,\hat{\pi}(x)} + \frac{(\hat{\mu}_0^A(x) - \mu_0^A(x))\,\hat{\tau}_{\text{init}}(x)\,(\pi(x) - \hat{\pi}(x))}{\hat{\delta}_A(x)\,\hat{\pi}(x)\,(1 - \hat{\pi}(x))}
\end{aligned}
$$
(16)

$$
\begin{aligned}
= {}& \frac{(\pi(x) - \hat{\pi}(x))}{\hat{\delta}_A(x)\,\hat{\pi}(x)\,(1 - \hat{\pi}(x))} \left[(\mu_0^Y(x) - \hat{\mu}_0^Y(x)) + (\hat{\mu}_0^A(x) - \mu_0^A(x))\,\hat{\tau}_{\text{init}}(x)\right] \\
& + (\hat{\tau}_{\text{init}}(x) - \tau(x)) + \frac{\pi(x)\delta_A(x)}{\hat{\pi}(x)\hat{\delta}_A(x)} \left(\tau(x) - \hat{\tau}_{\text{init}}(x)\right)
\end{aligned}
$$
(17)

$$
\begin{aligned}
= {}& \frac{(\pi(x) - \hat{\pi}(x))}{\hat{\delta}_A(x)\,\hat{\pi}(x)\,(1 - \hat{\pi}(x))} \left[(\mu_0^Y(x) - \hat{\mu}_0^Y(x)) + (\hat{\mu}_0^A(x) - \mu_0^A(x))\,\hat{\tau}_{\text{init}}(x)\right] \\
& + (\tau(x) - \hat{\tau}_{\text{init}}(x)) \left(\delta_A(x) - \hat{\delta}_A(x)\right) \pi(x) + (\tau(x) - \hat{\tau}_{\text{init}}(x))\,(\pi(x) - \hat{\pi}(x))\,\hat{\delta}_A(x).
\end{aligned}
$$
(18)

32   Applying the inequality $(a + b)^2 \leq 2(a^2 + b^2)$ together with Assumption 4 and the fact that $\pi(x) \leq 1$
33   yields

$$
\hat{r}(x)^2 \leq \frac{4}{\epsilon^4 \rho^2}\,(\pi(x) - \hat{\pi}(x))^2 \left[(\mu_0^Y(x) - \hat{\mu}_0^Y(x))^2 + (\hat{\mu}_0^A(x) - \mu_0^A(x))^2\,K^2\right]
$$
$$
+ 4\,(\tau(x) - \hat{\tau}_{\text{init}}(x))^2 \left(\delta_A(x) - \hat{\delta}_A(x)\right)^2 + 4\,(\tau(x) - \hat{\tau}_{\text{init}}(x))^2\,(\pi(x) - \hat{\pi}(x))^2. \quad (19)
$$

34 By setting $\widetilde{K} = \max\{K, 1\}$, we obtain

$$
\hat{r}(x)^2 \leq \frac{4\widetilde{K}^2}{\epsilon^4 \rho^2} \left( (\pi(x) - \hat{\pi}(x))^2 \left[ \left( \mu_0^Y(x) - \hat{\mu}_0^Y(x) \right)^2 + \left( \hat{\mu}_0^A(x) - \mu_0^A(x) \right)^2 + \left( \hat{\tau}_{\text{init}}(x) - \tau(x) \right)^2 \right] \right.
$$
$$
\left. + (\tau(x) - \hat{\tau}_{\text{init}}(x))^2 \left( \delta_A(x) - \hat{\delta}_A(x) \right)^2 \right). \tag{20}
$$

35 Applying expectations on both sides yields

$$
\mathbb{E}\left[ (\hat{\tau}_{\text{init}}(x) - \tau(x))^2 \right] \tag{21}
$$
$$
\lesssim \mathcal{R}(x) + \mathbb{E}\left[ (\hat{\tau}_{\text{init}}(x) - \tau(x))^2 \right] \left( \mathbb{E}\left[ \left( \hat{\delta}_A(x) - \delta_A(x) \right)^2 \right] + \mathbb{E}\left[ (\hat{\pi}(x) - \pi(x))^2 \right] \right)
$$
$$
+ \mathbb{E}\left[ (\hat{\pi}(x) - \pi(x))^2 \right] \left( \mathbb{E}\left[ \left( \hat{\mu}_0^Y(x) - \mu_0^Y(x) \right)^2 \right] + \mathbb{E}\left[ \left( \hat{\mu}_0^A(x) - \mu_0^A(x) \right)^2 \right] \right), \tag{22}
$$

36 because $(\hat{\pi}(x), \hat{\delta}_A(x)) \perp\!\!\!\perp (\hat{\mu}_0^Y(x), \hat{\mu}_0^A(x), \hat{\tau}_{\text{init}}(x))$ due to sample splitting. The claim follows now
37 by applying Assumption 3. □

## A.3  Proof of Theorem 3 (Convergence rate of the Wald estimator)

39 *Proof.* We define $\widetilde{C} = \max\{C, 1\}$ and obtain the upper bound

$$
(\hat{\tau}_W(x) - \tau(x))^2 \tag{23}
$$
$$
= \left( \frac{(\hat{\mu}_1^Y(x) - \mu_1^Y(x))\, \delta_A(x) + (\mu_0^Y(x) - \hat{\mu}_0^Y(x))\, \delta_A(x) + (\delta_A(x) - \hat{\delta}_A(x))\, \delta_Y(x)}{\delta_A(x)\, \hat{\delta}_A(x)} \right)^2 \tag{24}
$$
$$
\leq \frac{4\widetilde{C}^2}{\rho^2 \widetilde{\rho}^2} \left[ (\hat{\mu}_1^Y(x) - \mu_1^Y(x))^2 + (\hat{\mu}_0^Y(x) - \mu_0^Y(x))^2 + (\delta_A(x) - \hat{\delta}_A(x))^2 \right] \tag{25}
$$
$$
\leq \frac{8\widetilde{C}^2}{\rho^2 \widetilde{\rho}^2} \left[ (\hat{\mu}_1^Y(x) - \mu_1^Y(x))^2 + (\hat{\mu}_0^Y(x) - \mu_0^Y(x))^2 + (\hat{\mu}_1^A(x) - \mu_1^A(x))^2 \right.
$$
$$
\left. + (\hat{\mu}_0^A(x) - \mu_0^A(x))^2 \right], \tag{26}
$$

40 where we used the inequality $(a + b)^2 \leq 2(a^2 + b^2)$ several times. Taking expectations and applying
41 the smoothness assumptions yields the result. □

## B Theoretical analysis under sparsity assumptions

In Sec. 4.2, we analyzed MRIV theoretically by imposing smoothness assumptions on the underlying data generating process. In particular, we derived a multiple robust convergence rate and showed that MRIV outperforms the Wald estimator if the oracle ITE is smoother than its components. In this section, we derive similar results by relying on a different set of assumptions. Instead of using smoothness, we make assumptions on the level of sparsity of the ITE components. This assumption is often imposed in high-dimensional settings ($n < p$) and is in line with previous literature on analyzing ITE estimators [4, 8].

In the following, we say a function $f(x)$ is $k$-sparse, if it is linear in $x \in \mathbb{R}^p$ and it only depends on $k < \min\{n, p\}$ predictors. [22] showed, that in this case the minimax rate of $f(x)$ is given by $\frac{k \log(p)}{n}$. The linearity assumption can be relaxed to an additive structural assumption, which we omit here for simplicity. In the following, we replace the smoothness conditions in Assumption 3 with sparsity conditions.

**Assumption 6** (Sparsity). We assume that (1) the nuisance components $\mu_i^Y(\cdot)$ are $\alpha$-sparse, $\mu_i^A(\cdot)$ and $\delta_A(\cdot)$ are $\beta$-sparse, and $\pi(\cdot)$ is $\delta$-sparse; (2) all nuisance components are estimated with their respective minimax rate of $\frac{k \log(p)}{n}$, where $k \in \{\alpha, \beta, \delta\}$; and (3) the oracle ITE $\tau(\cdot)$ is $\gamma$-sparse and the initial ITE estimator $\hat{\tau}_{\mathrm{init}}$ converges with rate $r_\tau(n)$.

We restate now our result from Theorem 3 for MRIV using the sparsity assumption.

**Theorem 5** (MRIV upper bound under sparsity). *We consider the same setting as in Theorem 2 under the sparsity assumption 6. If the second-stage estimator $\hat{\mathbb{E}}_n$ yields the minimax rate $\frac{\gamma \log(p)}{n}$ and satisfies Assumption 5, the oracle risk is upper bounded by*

$$\mathbb{E}\left[(\hat{\tau}_{\mathrm{MRIV}}(x) - \tau(x))^2\right] \lesssim \frac{\gamma \log(p)}{n} + r_\tau(n) \frac{(\beta + \delta) \log(p)}{n} + \frac{(\alpha + \beta)\delta \log^2(p)}{n^2}.$$

*Proof.* Follows immediately from the proof of Theorem 2, i.e., from Eq.(21) by applying Ass- 6. □

Again, we obtain a multiple robust convergence rate for MRIV in the sense that MRIV achieves a fast rate even if the initial estimator or several nuisance estimators converge slowly. More precisely, for a fast convergence rate of $\hat{\tau}_{\mathrm{MRIV}}(x)$, it is sufficient if either: (1) $r_\tau(n)$ decreases fast and $\delta$ is small; (2) $r_\tau(n)$ decreases fast and $\alpha$ and $\beta$ are small; or (3) all $\alpha$, $\beta$, and $\delta$ are small.

We derive now the corresponding rate for the Wald estimator.

**Theorem 6** (Wald oracle upper bound). *Given estimators $\hat{\mu}_i^Y(x)$ and $\hat{\mu}_i^A(x)$. Let $\hat{\delta}_A(x) = \hat{\mu}_1^A(x) - \hat{\mu}_0^A(x)$ satisfy Assumption 4. Then, under Assumption 6 the oracle risk of the Wald estimator $\hat{\tau}_W(x)$ is bounded by*

$$\mathbb{E}\left[(\hat{\tau}_W(x) - \tau(x))^2\right] \lesssim \frac{(\alpha + \beta) \log(p)}{n} \tag{27}$$

*Proof.* Follows immediately from the proof of Theorem 3, i.e., from Eq.(23) by applying Ass- 6. □

If $\alpha = \beta = \delta$, we obtain the rates

$$\mathbb{E}\left[(\hat{\tau}_{\mathrm{MRIV}}(x) - \tau(x))^2\right] \lesssim \frac{\gamma \log(p)}{n} + \frac{\alpha^2 \log^2(p)}{n^2} \quad \text{and} \quad \mathbb{E}\left[(\hat{\tau}_W(x) - \tau(x))^2\right] \lesssim \frac{\alpha \log(p)}{n}, \tag{28}$$

which means that $\hat{\tau}_{\mathrm{MRIV}}(x)$ outperforms $\hat{\tau}_W(x)$ for $\gamma < \alpha$, i.e., if the oracle ITE is more sparse than its components.

 # C   Simulated data

77  In the following, we describe how we simulate synthetic data for the experiments in Sec. 5.1 from the
78  main paper. As mentioned therein, we simulate the ITE components from Gaussian processes using
79  the prior induced by the Matern kernel [12]

$$K_{\ell,\nu}(x_i, x_j) = \frac{1}{\Gamma(\nu)2^{\nu-1}} \left( \frac{\sqrt{2\nu}}{\ell} \|x_i - x_j\|_2 \right)^\nu K_\nu \left( \frac{\sqrt{2\nu}}{\ell} \|x_i - x_j\|_2 \right), \tag{29}$$

80  where $\Gamma(\cdot)$ is the Gamma function and $K_\nu(\cdot)$ is the modified Bessel function of second kind. Here, $\ell$
81  is the length scale of the kernel and $\nu$ controls the smoothness of the sampled functions.

82  We set $\ell = 1$ and sample functions $\delta_Y \sim \mathcal{GP}(0, K_{\ell,\gamma})$, $\mu_0^Y \sim \mathcal{GP}(0, K_{\ell,\alpha})$, $f_1 \sim \mathcal{GP}(0, K_{\ell,\beta})$,
83  $f_0 \sim \mathcal{GP}(0, K_{\ell,\beta})$ and $g \sim \mathcal{GP}(0, K_{\ell,\beta})$. Then, we define $\mu_1^Y = \delta_Y + \mu_0^Y$, $\mu_1^A = 0.3 \cdot \sigma \circ f_1 + 0.7$,
84  $\mu_0^A = 0.3 \cdot \sigma \circ f_0$, $\delta_A = \mu_1^A - \mu_0^A$, $\mu_0^Y = c_0 \delta_A$, and $\pi = \sigma \circ g$. Finally, we set the oracle ITE to

$$\tau = \frac{\mu_1^Y - \mu_0^Y}{\mu_1^A - \mu_0^A} = \frac{\delta_Y}{\delta_A}. \tag{30}$$

85  Note that we can create a setup where the ITE $\tau$ is smoother than its components by using a small
86  $\alpha/\beta$ ratio. An example is shown in Fig. 1.

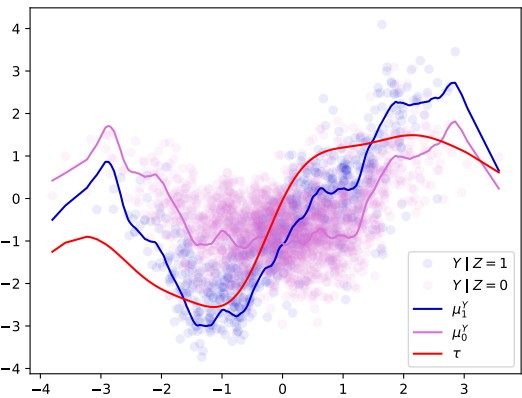

Figure 1: Gaussian process simulation for $\alpha = 1.5$ and $\beta = 50$.

87  In the following, we describe how we generate data the $(X, Z, A, Y)$ using the ITE components
88  $\mu_i^Y(x)$, $\mu_i^A(x)$, and $\pi(x)$. We begin by sampling $n$ observed confounder $X \sim \mathcal{N}(0, 1)$, unobserved
89  confounders $U \sim \mathcal{N}(0, 0.2^2)$, and instruments $Z \sim \mathrm{Bernoulli}(\pi(X))$. Then, we obtain treatments
90  via

$$A = Z \, \mathbb{1}\{U + \epsilon_A > \alpha_1(X)\} + (1 - Z) \, \mathbb{1}\{U + \epsilon_A > \alpha_0(X)\} \tag{31}$$

91  with indicator function $\mathbb{1}$, noise $\epsilon_A \sim \mathcal{N}(0, 0.1^2)$, and $\alpha_i(X) = \Phi^{-1}\left(1 - \mu_i^A(X)\right)\sqrt{0.1^2 + 0.2^2}$,
92  where $\Phi^{-1}$ denotes the quantile function of the standard normal distribution. Finally, we generate the
93  outcomes via

$$Y = A \left( \frac{(\mu_1^A(X) - 1)\mu_0^Y(X) - \mu_0^A(X)\mu_1^Y(X) + \mu_1^Y(X)}{\delta_A(X)} \right) \tag{32}$$

$$+ (1 - A) \left( \frac{\mu_1^A(X)\mu_0^Y(X) - \mu_0^A(X)\mu_1^Y(X)}{\delta_A(X)} \right) + \alpha_U U + \epsilon_Y, \tag{33}$$

94  where $\epsilon_Y \sim \mathcal{N}(0, 0.3^2)$ is noise and $\alpha_U > 0$ is a parameter indicating the level of unobserved
95  confounding. This choice of $A$ and $Y$ in Eq. (31) and Eq. (32), respectively, implies that $\tau(x)$ is
96  indeed the ITE, i.e., it holds that $\tau(x) = \mathbb{E}[Y(1) - Y(0) \mid X = x]$.

97 **Lemma 7.** *Let $(X, Z, A, Y)$ be sampled from the the previously described procedure. Then, it holds*
98 *that*

$$\mu_i^A(x) = \mathbb{E}[A \mid Z = i, X = x] \quad and \quad \mu_i^Y(x) = \mathbb{E}[Y \mid Z = i, X = x]. \tag{34}$$

99 *Proof.* The first claim follows from

$$\mathbb{E}[A \mid Z = i, X = x] = \mathbb{P}\left(U + \epsilon_A > \alpha_i(x)\right) = 1 - \Phi(\Phi^{-1}(1 - \mu_i^A(x))) = \mu_i^A(x), \tag{35}$$

100 because $U + \epsilon_A \sim \mathcal{N}(0, \sqrt{0.1^2 + 0.2^2})$. The second claim follows from

$$\mathbb{E}[Y \mid Z = i, X = x] = \mu_i^A(x) \left( \frac{(\mu_1^A(x) - 1)\mu_0^Y(x) - \mu_0^A(x)\mu_1^Y(x) + \mu_1^Y(x)}{\delta_A(x)} \right) \tag{36}$$

$$+ (1 - \mu_i^A(x)) \left( \frac{\mu_1^A(x)\mu_0^Y(x) - \mu_0^A(x)\mu_1^Y(x)}{\delta_A(x)} \right) \tag{37}$$

$$= \frac{\mu_i^Y(x)\delta_A(x)}{\delta_A(x)} = \mu_i^Y(x). \tag{38}$$

101 $\qquad\qquad\qquad\qquad\qquad\qquad\qquad\qquad\qquad\qquad\qquad\qquad\qquad\qquad\qquad\qquad\qquad\qquad\qquad\quad\square$

## D    Oregon health insurance experiment

The so-called *Oregon health insurance experiment*[1] (OHIE) [6] was an important RCT with non-compliance. It was intentionally conducted as large-scale effort among public health to assess the effect of health insurance on several outcomes such as health or economic status. In 2008, a lottery draw offered low-income, uninsured adults in Oregon participation in a Medicaid program, providing health insurance. Individuals whose names were drawn could decide to sign up for the program.

In our analysis, the lottery assignment is the instrument $Z$, the decision to sign up for the Medicaid program is the treatment $A$, and an overall health score is the outcome $Y$. The outcome was obtained after a period of 12 months during in-person interviews. We use the following covariates $X$: age, gender, language, the number of emergency visits before the experiment, and the number of people the individual signed up with. The latter is used to control for peer effects, and it is important to include this variable in our analysis as it is the only variable influencing the propensity score (see below). We extract $\sim$ 10,000 observations from the OHIE data and plot the histograms of all variables in Fig. 2. We can clearly observe the presence of non-compliance within the data, because the ratio of treated / untreated individuals is much lower than the corresponding ratio for the treatment assignment.

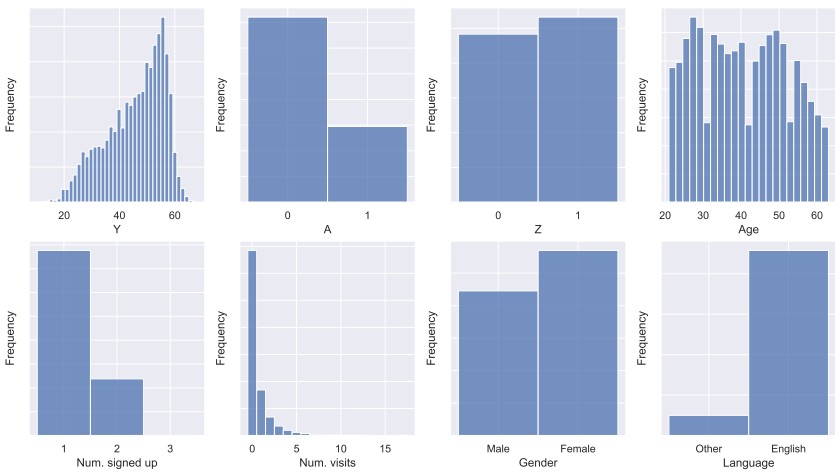

Figure 2: Histograms of each variable in our sample from OHIE.

The data collection in the OHIE was done follows: After excluding individuals below the age of 19, above the age of 64, and individuals with residence outside of Oregon, 74,922 individuals were considered for the lottery. Among those, 29,834 were selected randomly and were offered participation in the program. However, the probability of selection depended on the number of household members on the waiting list: for instance, an individual who signed up with another person was twice as likely to be selected. From the 74,922 individuals, 57,528 signed up alone, 17,236 signed up with another person, and 158 signed up with two more people on the waiting list. Thus, the probability of being selected conditional on the number of household members on the waiting list follows the multivariate version of Wallenius' noncentral hypergeometric distribution [2].

**Propensity score:** We computed the propensity score as follows. To account for the Wallenius' noncentral hypergeometric distribution, we use the R package *BiasedUrn* to calculate the propensity score $\pi(x) = \mathbb{P}(Z = 1 \mid X = x)$. We obtained

$$\pi(x) = \begin{cases} 0.345, & \text{if individual } x \text{ signed up alone,} \\ 0.571, & \text{if individual } x \text{ signed up with one more person,} \\ 0.719, & \text{if individual } x \text{ signed up with two more people.} \end{cases} \tag{39}$$

During the training of both MRIV and DRIV, we use the calculated values from Eq. (39) for the propensity score.

---

[1]Data available here: https://www.nber.org/programs-projects/projects-and-centers/oregon-health-insurance-experiment

## E Details for baseline methods

In this section, we give a brief overview on the baselines which we used in our experiments. We implemented: (1) ITE methods for unconfoundedness [8, 13]; (2) general IV methods, i.e., IV methods developed for IV settings with multiple or continuous instruments and treatments [1, 7, 14, 15, 20, 21]; and (3) two instantiations of the Wald estimator for the binary IV setting [16].

### E.1 ITE methods for unconfoundedness

Many ITE methods assume *unconfoundedness*, i.e., that all confounders are observed in the data. Formally, the unconfoundedness assumption can be expressed in the potential outcomes framework as

$$Y(1), Y(0) \perp\!\!\!\perp A \mid X. \tag{40}$$

Under unconfoundedness, the ITE is identified as

$$\tau(x) = \mu_1(x) - \mu_0(x) \quad \text{with} \quad \mu_i(x) = \mathbb{E}[Y \mid A = i, X = x]. \tag{41}$$

Methods that assume unconfoundedness proceed by estimating $\mu_i(x) = \mathbb{E}[Y \mid A = i, X = x]$ from Eq. (41). However, if unobserved confounders $U$ exist, it follows that

$$\tau(x) = \mathbb{E}[Y \mid A = 1, X = x, U] - \mathbb{E}[Y \mid A = 0, X = x, U] \neq \mu_1(x) - \mu_0(x), \tag{42}$$

which means that estimators that assume unconfoundedness are generally biased. Nevertheless, we include two baselines that assume unconfoundedness into our experiments: TARNet [13] and the DR-learner [8].

**TARNet** [13]: TARNet [13] is a neural network that estimates the ITE components $\mu_i(x)$ from Eq. 41 by learning a shared representation $\Phi(x)$ and two potential outcome heads $h_i(\Phi(x))$. We train TARNet by minimizing the loss

$$\mathcal{L}(\theta) = \sum_{i=1}^{n} L\left(h_{a_i}(\Phi(x_i, \theta_\Phi), \theta_{h_i}), y_i\right), \tag{43}$$

where $\theta = (\theta_{h_1}, \theta_{h_0}, \theta_\Phi)$ denotes the model parameters and $L$ denotes squared loss if $Y$ is continuous or binary cross entropy loss if $Y$ is binary.

*Note regarding balanced representations:* In [13], the authors propose to add an additional regularization term inspired from domain adaptation literature, which forces TARNet to learn a balanced representation $\Phi(x)$, i.e., that minimizes the distance the treatment and control group in the feature space. They showed that this approach leads to minimization of a generalization bound on the ITE estimation error if the representation is invertible.

In our experiments, we refrained from learning balanced representations because minimizing the regularized loss from [13] does not necessarily result in an invertible representation and thus may even harm the estimation performance. For a detailed discussion, we refer to [4]. Furthermore, by leaving out the regularization, we ensure comparability between the different baselines. If balanced representations are desired, the balanced representation approach could also be extended to MRIV-Net, as we also build MRIV-Net on learning shared representations.

**DR-learner** [8]: The DR-learner [8] is a meta learner that takes arbitrary estimators of the ITE componenets $\mu_i$ and the propensity score $\pi(x) = \mathbb{P}(A = 1 \mid X = x)$ as input and performs a pseudo outcome regression by using the pseudo outcome

$$\hat{Y}_0 = \left(\frac{A}{\hat{\pi}(X)} - \frac{1 - A}{1 - \hat{\pi}(X)}\right) Y + \left(1 - \frac{A}{\hat{\pi}(X)}\right) \hat{\mu}_1(X) - \left(1 - \frac{1 - A}{1 - \hat{\pi}(X)}\right) \hat{\mu}_0(X). \tag{44}$$

In our experiments, we use TARNet as base method to provide initial estimators $\hat{\mu}_i(X)$. We further learn propensity score estimates $\hat{\pi}(X)$ by adding a seperate representation to TARNet as done in [13].

### E.2 General IV methods

**2SLS** [20]: 2SLS [20] is a linear two-stage approach. First, the treatments $A$ are regressed on the instruments $Z$ and fitted values $\hat{A}$ are obtained. In the second stage, the outcome $Y$ is regressed on $\hat{A}$. We implement 2SLS using the scikit-learn package.

**KIV** [14]: Kernel IV [14] generalizes 2SLS to nonlinear settings. KIV assumes that the data is generated by

$$Y = f(A) + U, \tag{45}$$

where $U$ is an additive unobserved confounder and $f$ is some unknown (potentially nonlinear) structural function. KIV then models the structural function via

$$f(a) = \mu^t \psi(a) \quad \text{and} \quad \mathbb{E}[\psi(A) \mid Z = z] = V\phi(z), \tag{46}$$

where $\psi$ and $\phi$ are feature maps. Here, kernel ridge regressions instead of linear regressions are used in both stages to estimate $\mu$ and $V$.

Following [14] we use the exponential kernel [12] and set the length scale to the median inter-point distance. KIV does not provide a direct way to incorporate the observed confounders $X$. Hence, we augment both the instrument and the treatment with $X$, which is consistent with previous work [1, 21]. We also use two different samples for each stage as recommended in [14].

**DFIV** [21]: DFIV [21] is a similar approach KIV in generalizing 2SLS to nonlinear setting by assuming Eq. (45) and Eq. (46). However, instead of using kernel methods, DFIV models the features maps $\psi_{\theta_A}$ and $\phi_{\theta_Z}$ as neural networks with parameters $\theta_A$ and $\theta_Z$, respectively. DFIV is trained by iteratively updating the parameters $\theta_A$ and $\theta_Z$. The authors also provide a training algorithm that incorporates observed confounders $X$, which we implemented for our experiments. During training, we used two different datasets for each of the two stages as described in in the paper.

**DeepIV** [7]: DeepIV [7] also assumes additive unobserved confounding as in Eq. (45), but leverages the identification result [10]

$$\mathbb{E}[Y \mid X = x, Z = z] = \int h(a, x) \, dF(a \mid x, z), \tag{47}$$

where $h(a, x) = f(a, x) + \mathbb{E}[U \mid X = x]$ is the target counterfactual prediction function. DeepIV estimates $F(a \mid x, z)$, i.e., the conditional distribution function of the treatment $A$ given observed covariates $X$ and instruments $Z$, by using neural networks. Because we consider only binary treatments, we simply implement a (tunable) feed-forward neural network with sigmoid activation function. Then, DeepIV proceeds by learning a second stage neural network to solve the inverse problem defined by Eq. (47).

**DeepGMM** [1]: DeepGMM [1] adopts neural networks for IV estimation inspired by the (optimally weighted) Generalized Method of Moments. The DeepGMM estimator is defined as the solution of the following minimax game:

$$\hat{\theta} \in \arg\min_{\theta \in \Theta} \sup_{\tau \in T} \frac{1}{n} \sum_{i=1}^{n} f(z_i, \tau)(y_i - g(a_i, \theta)) - \frac{1}{4n} \sum_{i=1}^{n} f^2(z_i, \tau)(y_i - g(a_i, \widetilde{\theta}))^2, \tag{48}$$

where $f(z_i, \cdot)$ and $g(a_i, \cdot)$ are parameterized by neural networks. As recommended in [1], we solve this optimization via adversarial training with the Optimistic Adam optimizer [5], where we set the parameter $\widetilde{\theta}$ to the previous value of $\theta$.

**DMLIV** [15]: DMLIV [15] assumes that the data is generated via

$$Y = \tau(X)A + f(X) + U, \tag{49}$$

where $\tau$ is the ITE $f$ some function of the observed covariates. First, DMLIV estimates the functions $q(X) = \mathbb{E}[Y \mid X]$, $h(Z, X) = \mathbb{E}[A \mid Z, X]$, and $p(X) = \mathbb{E}[A \mid X]$. Then, the ITE is learned by minimizing the loss

$$\mathcal{L}(\theta) = \sum_{i=1} (y_i - \hat{q}(x_i) - \hat{\tau}(x_i, \theta)(\hat{h}(z_i, x_i) - \hat{p}(x_i)))^2, \tag{50}$$

where $\hat{\tau}(X, \cdot)$ is some model for $\tau(X)$. In our experiments, we use (tunable) feed-forward neural networks for all estimators.

**DRIV** [15]: DRIV [15] is a meta learner, originally proposed in combination with DMLIV. It requires initial estimators for $q(X)$, $p(X)$, $\pi(X) = \mathbb{E}[Z \mid X = x]$, and $f(X) = \mathbb{E}[A \cdot Z \mid X = x]$ as well as an initial ITE estimatior $\hat{\tau}_{\text{init}}(X)$ (e.g., from DMLIV). The ITE is then estimated by a pseudo regression on the following doubly robust pseudo outcome:

$$\hat{Y}_{\text{DR}} = \hat{\tau}_{\text{init}}(X) + \frac{(Y - \hat{q}(X) - \hat{\tau}_{\text{init}}(X)(A - \hat{p}(X))Z - \hat{\pi}(X))}{\hat{f}(X) - \hat{p}(X)\hat{r}(X)}. \tag{51}$$

213 We implement all regressions using (tunable) feed-forward neural networks.

214 Comparison between DRIV vs. MRIV: There are two key differences between our paper and [15]:
215 (i) Our MRIV is multiply robust, while DRIV is only doubly robust. (ii) We derive a multiple robust
216 convergence rate, while the rate in [15] is not robust with respect to the nuisance rates.

217 Ad (i): Both MRIV and DRIV perform a pseudo-outcome regression on the efficient influence
218 function (EIF) of the ATE. The key difference: DRIV uses the doubly robust parametrization of the
219 EIF from [11], whereas we use the multiply robust parametrization of the EIF from [17] [2]. Hence,
220 our MRIV frameworks extends DRIV in a non-trivial way to achieve multiple robustness (rather
221 than doubly robustness). Thus, our estimator is consistent in the union of *three* different model
222 specifications rather than *two*.[3]

223 Ad (ii): Here, we compare the convergence rates from DRIV and our MRIV and, thereby, show the
224 strengths of our MRIV. To this end, let us assume that the pseudo regression function is $\gamma$-smooth and
225 that we use the same second-stage estimator $\hat{\mathbb{E}}_n$ with minimax rate $n^{-\frac{2\gamma}{2\gamma+p}}$ for both DRIV and MRIV.
226 If the nuisance parameters $q(X)$, $p(X)$, $f(X)$, and $\pi(X)$ are $\alpha$-smooth and further are estimated
227 with minimax rate $n^{\frac{-2\alpha}{2\alpha+p}}$, Corollary 4 from [15] states that DRIV converges with rate

$$\mathbb{E}\left[(\hat{\tau}_{\mathrm{DRIV}}(x) - \tau(x))^2\right] \lesssim n^{\frac{-2\gamma}{2\gamma+p}} + n^{\frac{-4\alpha}{2\alpha+p}}.$$

228 In contrast, MRIV assumes estimation of the nuisance parameters $\mu_0^Y(x)$ with rate $n^{\frac{-2\alpha}{2\alpha+p}}$, $\mu_0^A(x)$
229 and $\delta_A(x)$ with rate $n^{\frac{-2\beta}{2\beta+p}}$, and $\pi(x)$ with rate $n^{\frac{-2\delta}{2\delta+p}}$. If the initial estimator $\hat{\tau}_{\mathrm{init}}(x)$ converges with
230 rate $r_\tau(n)$, our Theorem 2 yields the rate

$$\mathbb{E}\left[(\hat{\tau}_{\mathrm{MRIV}}(x) - \tau(x))^2\right] \lesssim n^{\frac{-2\gamma}{2\gamma+p}} + r_\tau(n)\left(n^{\frac{-2\beta}{2\beta+p}} + n^{\frac{-2\delta}{2\delta+p}}\right) + n^{-2\left(\frac{\alpha}{2\alpha+p}+\frac{\delta}{2\delta+p}\right)} + n^{-2\left(\frac{\beta}{2\beta+p}+\frac{\delta}{2\delta+p}\right)}.$$

231 If all nuisance parameters converge with the same minimax rate of $n^{\frac{-2\alpha}{2\alpha+p}}$, the rates of DRIV and
232 our MRIV coincide. However, different to DRIV, our rate is additionally multiple robust in spirit of
233 Theorem 1. This presents a crucial strength of our MRIV over DRIV: For example, if $\delta$ is small (slow
234 convergence of $\hat{\pi}(x)$), our MRIV still with fast rate as long as $\alpha$ and $\beta$ are large (i.e., if the other
235 nuisance parameters are sufficiently smooth).

## E.3 Wald estimator

237 Finally, we consider the Wald estimator [16] for the binary IV setting. More precisely, we estimate
238 the ITE components $\mu_i^Y(x)$ and $\mu_i^A(x)$ seperately and plug them into

$$\tau(x) = \frac{\hat{\mu}_1^Y(x) - \hat{\mu}_0^Y(x)}{\hat{\mu}_1^A(x) - \hat{\mu}_0^A(x)}. \tag{52}$$

239 We consider two versions of the Wald estimator:

240 **Linear:** We use linear regressions to estimate the $\mu_i^Y(x)$ and logistic regressions to estimate the
241 $\mu_i^A(x)$.

242 **BART:** We use Bayesian additive regression trees [3] trees to estimate the $\mu_i^Y(x)$ and random forest
243 classifier to estimate the $\mu_i^A(x)$.

---

[2]For a detailed discussion on multiple robustness and the importance of the EIF parametrization, we refer to [18], Section 4.5.

[3]On a related note, a similar, important contribution of developing multiply robust method was recently made for the average treatment effect. Here, the estimator of [11] was extended by the estimator of [17] to allow for multi robustness. Yet, this different from our work in that it focuses on the average treatment effect, while we study the individual treatment effect in our paper.

## F Visualization of predicted ITEs

We plot the predicted ITEs for the different baselines and MRIV-Net in Fig. 3 (for $n = 3000$). As expected, the linear methods (2SLS and linear Wald) are not flexible enough to provide accurate ITE estimates. We also observe that the curve of MRIV-Net without MRIV is quite wiggly, i.e., the estimator has a relatively large variance. This variance is reduced when the full MRIV-Net is applied. As a result, curve is much smoother. This is reasonable because MRIV does not estimate the ITE components individually, but estimates the ITE directly via the Stage 2 pseudo outcome regression. Overall, this confirms the superiority of our proposed framework.

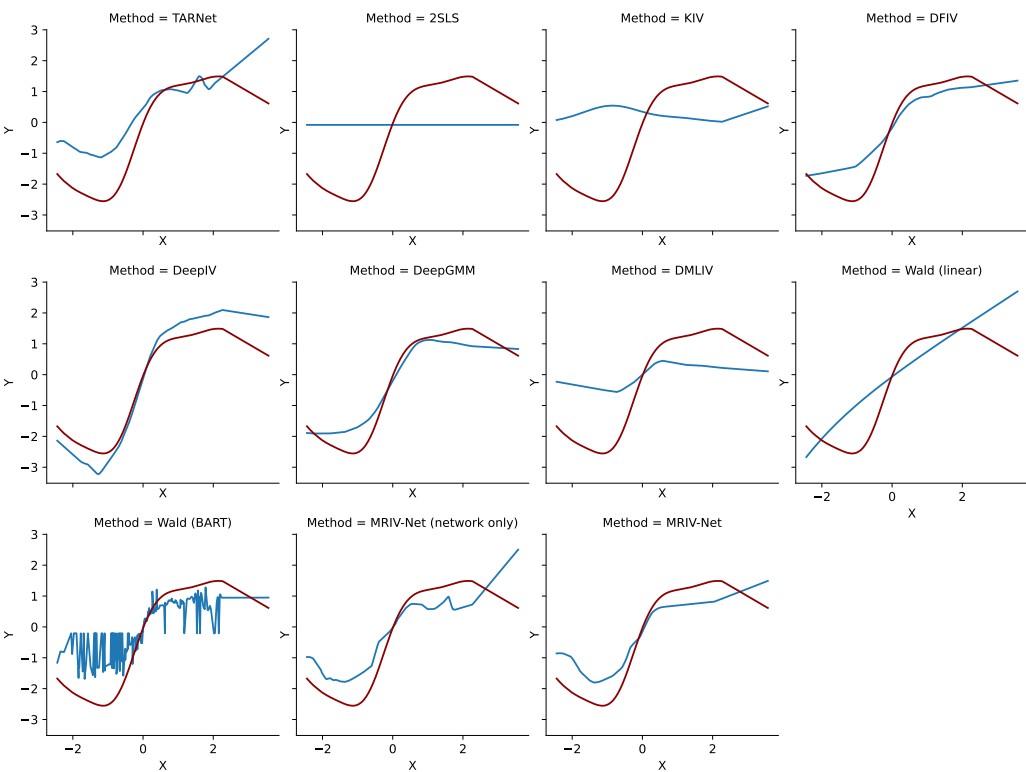

Figure 3: Predicted ITEs (blue) and oracle ITE (red) for different baselines.

## G Implementation details and hyperparameter tuning

**Implementation details for deep learning models:** To make the performance of the deep learning models comparable, we implemented all feed-forward neural networks (including MRIV-Net) as follows: We use two hidden layers with RELU activation functions. We also incorporated a dropout layer for each hidden layer. We trained all models with the Adam optimizer [9] using 100 epochs. Exceptions are only DFIV and DeepGMM, where we used 200 epochs for training, accounting for slower convergence of the respective (adversarial) training algorithms. For DeepGMM, we further used Optimistic Adam [5] as in the original paper.

**Training times:** We report the approximate times needed to train the deep learning models on our simulated data with $n = 5000$ in Table 1. For training, we used an AMD Ryzen Pro 7 CPU. Compared to DMLIV and DRIV, the training of MRIV-Net is faster because only a single neural network is trained.

Table 1: Training times for deep learning models (in seconds).

| TARNet | TARNet + DR | DFIV | DeepIV | DeepGMM | DMLIV | DMLIV + DRIV | MRIV-Net |
|--------|-------------|------|--------|---------|-------|--------------|----------|
| ∼10.62 | ∼28.57 | ∼164.98 | ∼30.21 | ∼17.31 | ∼74.98 | ∼91.12 | ∼32.20 |

**Hyperparameter tuning:** We performed hyperparameter tuning for all deep learning models (including MRIV-Net), KIV, and the BART Wald estimator on all datasets. For all methods except KIV and DFIV, we split the data into a training set (80%) and a validation set (20%). We then performed 40 random grid search iterations and chose the set of parameters that minimized the respective training loss on the validation set. In particular, the tuning procedure was the same for all baselines, which ensures that the performance gain of MRIV-Net is due to the method itself and not due to larger flexibility. Exceptions are only KIV and DFIV, for which we implemented the customized hyperparameter tuning algorithms proposed in [14] and [21] to ensure consistency with prior literature. For the meta learners (DR-learner, DRIV, and MRIV), we first performed hyperparameter tuning for the base methods and nuisance models, before tuning the pseudo outcome regression neural network by using the input from the tuned models. The tuning ranges for the hyperparameter are shown in Table 2. These include both the hyperparameter rangers shared across all neural networks and the model-specific hyperparameters. For reproducibility purposes, we publish the selected hyperparameters in our GitHub project as *.yaml* files.[4]

Table 2: Hyperparameter tuning ranges.

| MODEL | HYPERPARAMETER | TUNING RANGE |
|-------|----------------|--------------|
| Feed-forward neural networks (Shared parameter ranges for all deep learning baselines) | Hidden layer size(es) | $p, 5p, 10p, 20p, 30p$ (simulated data) $p, 3p, 5p, 8p, 10p$ (OHIE) |
| | Learning rate | 0.0001, 0.0005, 0.001, 0.005, 0.01 |
| | Batch size | 64, 128, 256 |
| | Dropout probability | 0, 0.1, 0.2, 0.3 |
| KIV | $\lambda$ (Ridge penalty first stage) | 5, 6, 7, 8, 9, 10, 12 |
| | $\xi$ (Ridge penalty second stage) | 5, 6, 7, 8, 9, 10, 12 |
| DFIV | $\lambda_1$ (Ridge penalty first stage) | 0.0001, 0.001, 0.01, 0.1 (simulated data) 0.01, 0.05, 0.1 (OHIE) |
| | $\lambda_2$ (Ridge penalty second stage) | 0.0001, 0.001, 0.01, 0.1 (simulated data) 0.01, 0.05, 0.1 (OHIE) |
| DeepGMM | $\lambda_f$ (learning rate multiplier) | 0.5, 1, 1.5, 2, 5 |
| Wald (BART) | Number of trees (BART) | 20, 30, 40, 50 |
| | Number of trees (Random forest classifier) | 20, 30, 40, 50 |

$p$ = network input size

**Hyperparameter robustness checks:** We also investigate the robustness of MRIV-Net with respect to hyperparameter choice. To To this, we fix the optimal hyperparameter constellation for our simulated data for $n = 3000$ and perturb the hidden layer sizes, learning rate, dropout probability, and batch size.

---

[4]Codes are in the supplementary materials. Codes are also available at https://anonymous.4open.science/r/MRIV-Net-0AC4 (Upon acceptance, we replace the link and point to a public GitHub repository).

The results are shown in Fig. 4. We observe that the RMSE only changes marginally when perturbing the different hyperparameters, indicating that our method is to a certain degree robust against hyperparameter misspecification. Furthermore, our results indicate that the performance improvement of MRIV-Net over the baselines observed in our experiments is not due to hyperparameter tuning, but to our method itself.

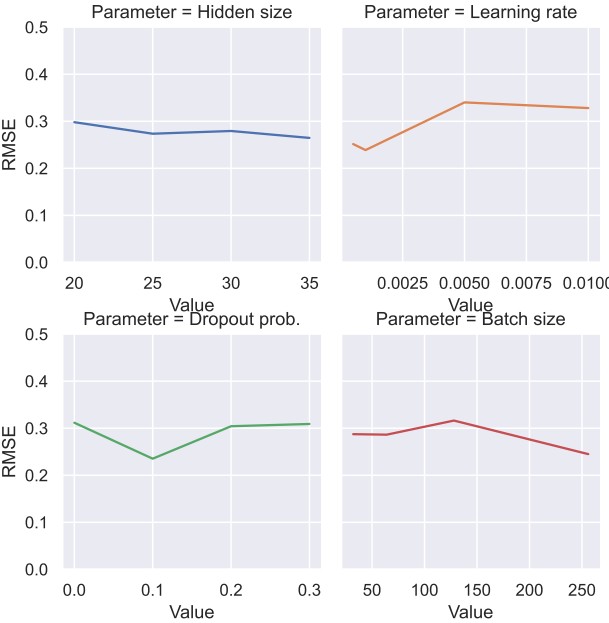

Figure 4: Robustness checks for different hyperparameters of MRIV-Net.

## H   Results for semi-synthetic data

In the main paper, we evaluated MRIV-Net both on synthetic and real-world data. Here, we provide additional results by constructing a semi-synthetic dataset on the basis of OHIE. It is common practice in causal inference literature to use semi-synthetic data for evaluation, because it combines advantages of both synthetic and real-world data. On the one hand, the real-world data part ensures that the data distribution is realistic and matches those in practice. On the other hand, the counterfactual ground-truth is still available, which makes it possible to measure the performance of ITE methods.

We construct our semi-synthetic data as follows: First, we extract the covariates $X \in \mathbb{R}^5$ and instruments $Z \in \{0, 1\}$ of our OHIE dataset from Sec. D. Then, we construct the treatment components $\mu_i^A(x)$ via

$$\mu_1^A(X) = 0.3 \cdot \sigma(X_1) + 0.7 \quad \text{and } \mu_0^A(X) = 0.3 \cdot \sigma(X_1), \tag{53}$$

where $X_1$ is the (standardized) age and $\sigma(\cdot)$ is the sigmoid function. The outcome components are constructed via

$$\mu_1^Y(X) = 0.5X_1^2 + \sum_{i=2}^{5} X_i^2 \quad \text{and } \mu_0^Y(X) = -0.5X_1^2 + \sum_{i=2}^{5} X_i^2. \tag{54}$$

We then sample treatments $A$ and outcomes $Y$ as in Eq. (31) and Eq. (32). Lemma 7 ensures that $\mu_i^Y(X) = \mathbb{E}[Y \mid Z = i, X]$ and $\mu_i^A(X) = \mathbb{E}[A \mid Z = i, X]$.

Given the above, the oracle ITE becomes

$$\tau(X) = \frac{X_1^2}{0.7}. \tag{55}$$

Note that $\tau(X)$ is sparse in the sense that it only depends on age, while the outcome components depend on all five covariates. Following our theoretical analysis in Sec. B, MRIV-Net should thus outperform methods that aim at estimating the components directly. This is confirmed in Table 3, where we show the results for all baselines and MRIV-Net on the semi-synthetic data. Indeed, we observe that MRIV-Net outperforms all other baselines, confirming both the superiority of our method as well as our theoretical results under sparsity assumptions from Sec. B.

Table 3: Results for semi-synthetic data.

| Method | $n = 3000$ | $n = 5000$ | $n = 8000$ |
|---|---|---|---|
| (1) STANDARD ITE | | | |
| TARNet [13] | $1.66 \pm 0.11$ | $1.58 \pm 0.07$ | $1.57 \pm 0.11$ |
| TARNet + DR [13, 8] | $1.31 \pm 0.28$ | $1.22 \pm 0.37$ | $1.12 \pm 0.15$ |
| (2) GENERAL IV | | | |
| 2SLS [19] | $1.34 \pm 0.06$ | $1.31 \pm 0.03$ | $1.32 \pm 0.02$ |
| KIV [14] | $1.97 \pm 0.10$ | $1.92 \pm 0.05$ | $1.93 \pm 0.05$ |
| DFIV [21] | $1.67 \pm 0.44$ | $1.63 \pm 0.47$ | $1.45 \pm 0.17$ |
| DeepIV [7] | $1.24 \pm 0.26$ | $0.99 \pm 0.22$ | $0.84 \pm 0.19$ |
| DeepGMM [1] | $1.39 \pm 0.03$ | $1.37 \pm 0.16$ | $1.18 \pm 0.16$ |
| DMLIV [15] | $2.12 \pm 0.10$ | $2.09 \pm 0.09$ | $2.02 \pm 0.11$ |
| DMLIV + DRIV [15] | $1.22 \pm 0.10$ | $1.18 \pm 0.19$ | $1.00 \pm 0.08$ |
| (3) WALD ESTIMATOR [16] | | | |
| Linear | $1.42 \pm 0.24$ | $1.28 \pm 0.07$ | $1.32 \pm 0.07$ |
| BART | $1.48 \pm 0.24$ | $1.29 \pm 0.04$ | $1.06 \pm 0.13$ |
| MRIV-Net (network only) | $1.11 \pm 0.15$ | $0.84 \pm 0.14$ | $0.95 \pm 0.21$ |
| MRIV-Net (ours) | $\mathbf{0.71 \pm 0.24}$ | $\mathbf{0.75 \pm 0.18}$ | $\mathbf{0.78 \pm 0.26}$ |

Reported: RMSE (mean $\pm$ standard deviation). Lower = better (best in bold)

# I  Results for cross-fitting

Here, we repeat our experiments from the main paper but now make use of *cross-fitting*. Recall that, in Theorem 2, we assume that the nuisance parameter estimation and the pseudo-outcome regression are performed on three independent samples. We now address this through *cross-fitting*. To this end, our aim is to show that our proposed MRIV framework is again superior.

For MRIV, we proceeded as follows: We split the sample $\mathcal{D}$ into three equally sized samples $\mathcal{D}_1$, $\mathcal{D}_2$, and $\mathcal{D}_3$. We then trained $\hat{\tau}_{init}(x)$, $\hat{\mu}_0^Y(x)$, and $\hat{\mu}_0^A(x)$ on $\mathcal{D}_1$, $\hat{\delta}_A(x)$ and $\hat{\pi}(x)$ on $\mathcal{D}_2$, and performed the pseudo-outcome regression on $\mathcal{D}_3$. Then, we repeated the same training procedure two times, but performed the pseudo-outcome regression on $\mathcal{D}_2$ and $\mathcal{D}_1$. Finally, we averaged the resulting three ITE estimators. For DRIV, we implemented the cross-fitting procedure described in [15]. For the DR-learner, we followed [8].

The results are in Table H. Importantly, the results confirm the effectiveness of our proposed MRIV. Overall, we find that our proposed MRIV outperforms DRIV for the vast majority of base methods when performing cross-fitting. Furthermore, MRIV-Net is highly competitive even when comparing it with the cross-fitted estimators. This shows that our heuristic to learn separate representations instead of performing sample splits works in practice. In sum, the results confirm empirically that our MRIV is superior.

Table 4: Results for base methods with different meta-learners (i.e., DRIV, and our MRIV) using cross-fitting and results for MRIV-Net without cross-fitting.

| Base methods \ Meta-learners | $n = 3000$ | | $n = 5000$ | | $n = 8000$ | |
|---|---|---|---|---|---|---|
| | DRIV | MRIV (ours) | DRIV | MRIV (ours) | DRIV | MRIV (ours) |
| **(1) STANDARD ITE** | | | | | | |
| TARNet [13] | **0.30 ± 0.02** | 0.36 ± 0.16 | 0.18 ± 0.06 | **0.16 ± 0.03** | 0.21 ± 0.08 | **0.13 ± 0.04** |
| TARNet + DR-learner [13, 8] | 0.85 ± 0.11 | | 0.66 ± 0.08 | | 0.67 ± 0.12 | |
| **(2) GENERAL IV** | | | | | | |
| 2SLS [19] | 0.42 ± 0.11 | **0.33 ± 0.09** | **0.20 ± 0.07** | 0.23 ± 0.11 | 0.24 ± 0.10 | **0.14 ± 0.02** |
| KIV [14] | 0.47 ± 0.18 | **0.45 ± 0.15** | 0.20 ± 0.06 | **0.19 ± 0.08** | 0.22 ± 0.04 | **0.15 ± 0.03** |
| DFIV [21] | 0.35 ± 0.05 | **0.28 ± 0.09** | 0.22 ± 0.10 | **0.18 ± 0.08** | 0.24 ± 0.12 | **0.16 ± 0.04** |
| DeepIV [7] | **0.38 ± 0.09** | 0.44 ± 0.16 | 0.20 ± 0.07 | **0.19 ± 0.07** | 0.20 ± 0.08 | **0.12 ± 0.02** |
| DeepGMM [1] | **0.42 ± 0.09** | 0.42 ± 0.16 | **0.19 ± 0.04** | 0.19 ± 0.07 | 0.22 ± 0.06 | **0.13 ± 0.02** |
| DMLIV [15] | **0.44 ± 0.09** | 0.46 ± 0.16 | 0.21 ± 0.04 | **0.19 ± 0.07** | 0.21 ± 0.05 | **0.14 ± 0.02** |
| **(3) WALD ESTIMATOR [16]** | | | | | | |
| Linear | 0.47 ± 0.23 | **0.36 ± 0.12** | 0.24 ± 0.05 | **0.20 ± 0.08** | 0.22 ± 0.05 | **0.15 ± 0.02** |
| BART | 0.43 ± 0.12 | **0.39 ± 0.12** | 0.14 ± 0.05 | **0.13 ± 0.05** | 0.23 ± 0.08 | **0.15 ± 0.02** |
| MRIV-Net\w network only (ours) | 0.35 ± 0.12 | **0.26 ± 0.11** | 0.19 ± 0.13 | **0.15 ± 0.03** | 0.18 ± 0.08 | **0.13 ± 0.03** |

Reported: RMSE (mean ± standard deviation). Lower = better (best in bold)

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
