# OpenReview forum: "Estimating individual treatment effects under unobserved confounding using binary instruments"
_NeurIPS.cc/2022/Conference — NeurIPS 2022 Submitted_

### Official Review · Reviewer_Fvf9 · 2022-07-09

**Rating:** 6
**Confidence:** 3
**Soundness:** 3 good
**Presentation:** 4 excellent
**Contribution:** 3 good

**Summary:**

This paper presents a two-stage regression method: In stage 1, they estimates nuisance components $\{\pi(x), \mu^Y_0, \mu^A_0, \delta_A(x)\}$; In stage 2, given an arbitrary initial ITE estimator ${\tau}_{\text {init }}(x)$ and  nuisance estimates $\{\pi(x), \mu^Y_0, \mu^A_0, \delta_A(x)\}$  (obtained from stage 1), they define a pseudo outcome for ITE estimation.  Theoretically, the framework proposed in this paper yields multiple robust convergence rates.

----
I thank the authors for addressing my comments. I increase my score to 6.

**Questions:**

In Figure 1, is it necessary to assume that X and U are independent?

Lines 79-82: Does this article focus on observational data or experiments data?

$\hat{Y}_0$ is usually used to denote $\mathbb{E}[Y|T=0]$ instead of ITE $\tau$, which is confusing.

Typos: Line 157: “We then derive we derive …”

Maybe the author can give a brief conclusion and future exploration.

**Limitations:**

I do not foresee any major limitations and/or societal impacts.

**Strengths And Weaknesses:**

This paper builds upon the theoretical results and propose a deep neural network architecture called MRIV-Net for ITE estimation. However, the proof of the theory and Appendices A-G are missing, which I cannot find in the supplemental material. This makes me doubt the solidity of the method and results. I will adjust my score according to the authors’ responses to my comments.

**Strengths**

- This paper considers a very important and interesting problem in Treatment Effect and proposes a novel estimator using binary instruments.
- The paper is very well written and provides extensive evaluation and discussion/comparison of past work.
- Theoretically, I believe that Theorem 1 is correct, and MRIV has Multiple robustness properties. But I think it's necessary to give a brief description or simply express the process with an example:

(1) Assuming $\hat{\mu}_0^Y=\mu_0^Y, \hat{\mu}_0^A=\mu_0^A, \hat{\delta}_A=\delta_A, \hat{\pi}=0.5, \hat{\tau}=0$:

$\hat{Y}_0 \leftarrow\left(\frac{Z-(1-Z)}{\delta_A(X)}\right)\left({Y-{\mu}_0^Y(X)}\right)$;

(2) Assuming $\hat{\mu}_0^Y=0, \hat{\mu}_0^A=0, \hat{\delta}_A=\delta_A, \hat{\pi}=\pi, \text{and } \hat{\tau}=0$:

$\hat{Y}_0 \leftarrow \left(\frac{Z-(1-Z)}{\delta_A(X)}\right)\left(\frac{Y}{Z {\pi}(X)+(1-Z)(1-{\pi}(X))}\right)$;

(3) Assuming $\hat{\mu}_0^Y=0, \hat{\mu}_0^A=0, $ $\hat{\delta}_A=1, \hat{\pi}=\pi, \text{and } \hat{\tau}=\tau$:

$\hat{Y}_0 \leftarrow\left({Z-(1-Z)}\right)\left(\frac{Y-A \tau}{Z {\pi}(X)+(1-Z)(1-{\pi}(X))}\right)+{\tau}$.

$\mathbb{E}[\hat{Y}^0∣X=x]=τ$.

**Weaknesses**

- For Eq. (3), at least two models need to be correctly specified to guarantee robust learning, which is a stronger condition than robust learning. The latter only requires either the propensity model or the potential outcome estimation model to be specified accurately.
- The proof of the theory and Appendices A-G are missing, which I cannot find in the supplemental material. This makes me doubt the solidity of the method and results.
- This paper should introduce the concept of multiple robust in the Introduction Section, and roughly explain that the original problem is simplified to 5 nuisance estimators —— the solution is still robust even if some models are misspecified.

---

> ### Author Response · Authors · 2022-08-02
> **Response to Reviewer Fvf9**
>
> Thank you very much for your review. We took all of your suggestions at heart and improved our work accordingly:
> 1. We added the appendix with all proofs to the rebuttal PDF.
> 2. We revised our introduction and the literature section to clarify the difference between previously proposed doubly robust methods for IV settings and our proposed MRIV framework (see revised Sections 1 and 3). Therein, we state clearly that our ITE framework is the *first* that is multiply robust (whereas existing works are only doubly robust).
> 3. We discuss in greater depth that our MRIV framework is applicable to both observational data and experiments with non-compliance.
> We highlighted our improvements (and other changes) in red color in our revised paper (see rebuttal PDF). Moreover, we provide a detailed point-by-point response below.
>
> ### Supplement
> We are very sorry that the appendix PDF was not included in the ZIP folder. We must have made a mistake when uploading to OpenReview. We have now added the appendix to the rebuttal PDF. The rebuttal PDF contains the complete appendix including the proofs for all our theoretical claims (see Appendix A). We are confident that this confirms the correctness of our theoretical results. Moreover, our appendix contains: (1) an additional theoretical analysis under sparsity assumptions, (2) details of the data generating processes, implementations and hyperparameter tuning procedures; and (3) additional experimental results.
>
> ### Responses to "Strengths"
> We followed your suggestions and added (1) a refined explanation and (2) a new example to explain the multiple robustness statement of Theorem 1. We have added the new materials directly after Theorem 1.
>
> ### Responses to "Weaknesses"
> 1.  **Multiple robustness vs double robustness:** We improved our paper by clarifying the difference between multiply and doubly robust methods in several sections of our revised paper. To this end, we made substantial changes to our introduction, our literature section, and Appendix E (see rebuttal PDF). In the IV setting, an estimator is said to be "doubly robust" if it is consistent in the union of **two** model specifications (see Wang and Tchetgen Tchetgen (2018)). Well-specification of just propensity model or the outcome models is generally *not sufficient* for consistency of doubly robust IV estimators. This is because additional nuisance parameters need to be estimated. Because we use the multiply robust paramatrization of the efficient influence function (EIF) in Wang and Tchetgen Tchetgen (2018), our estimator is consistent in the union of **three** model specifications. To improve our paper, we proceeded as follows: (i) We added these details to our literature section (see revised Section 3). (ii) We clarify the difference between our proposed MRIV framework and previously proposed doubly robust methods for IV settings. (iii) We also refer to Wang and Tchetgen Tchetgen (2018), Section 4.5, for a detailed discussion on multiple robustness and the importance of the EIF parametrization.
>
> 2. We added the appendix to the rebuttal PDF. Our appendix includes the proofs for our theoretical claims (see Appendix A).
>
> 3. We followed your suggestion and added a new paragraph to the introduction, describing different steps and properties of our framework (see Section 1). Here, we specifically explain that the original problem is simplified to estimating 5 nuisance parameters and performing a pseudo outcome regression. We also introduce the concept of multiple robustness (in the sense of consistency under miss-specifications and convergence rates). Throughout our paper, we emphasize the salient differences between double vs. multiple robustness. Thereby, we state also more clearly how our work is different from the prior literature and why ours is novel.
>
> ### Responses to "Questions"
>
> - In Figure 1, $U$ and $X$ do not need to be independent. We clarified this by adding an arrow between both variables in Figure 1 of our rebuttal PDF. We also changed the caption, which now states that our framework is flexible and works regardless of whether $U$ and $X$ are dependent or independent.
>
> - Our proposed framework is applicable to data from the data generating process in Section 2. This includes (1) observational data and (2) RCTs with non-compliance. Importantly, the latter ensures that our framework is *also* applicable when the instrument assignment is randomized (no $X$-$Z$ arrow) or when the propensity score is known. Such settings are known as "randomized controlled trials with non-compliance". To address your question, we revised our paper accordingly and now elaborate more clearly when our framework is applicable (see revised Section 2).
>
> - We agree that the notation $\hat{Y_0}$ can be improved. To address this, we now use $\hat{Y}_{\mathrm{MR}}$ instead.
>
> - We fixed the typo.
>
> - We added a conclusion (see new Section 6). Our new conclusion also includes an outlook on future research directions.

---

### Official Review · Reviewer_biU1 · 2022-07-09

**Rating:** 6
**Confidence:** 3
**Soundness:** 3 good
**Presentation:** 3 good
**Contribution:** 3 good

**Summary:**

The authors propose an algorithm for estimating ITE for binary instrument and binary treatment. The procedure has three steps.
-implement some initial estimator of ITE
-construct pseudo-outcome as sum of the initial estimator and a term combining several nuisances
-regress the pseudo-outcome on covariates
For this algorithm, the authors prove excess risk that scales as the sum of an oracle rate and several product rates.

**Questions:**

Please address the items in Originality, Quality, and Clarity and I will increase the score

**Strengths And Weaknesses:**

Originality

I will raise the score if the authors improve the framing of the contribution in the context of previous work.

1. The pseudo-outcome technique is relatively popular in causal inference; see e.g. Kennedy et al.’s work on dose response (2019) which Semenova and Chernozhukov (2021) extend to ITE. This paper extends those techniques, so I would like them to be cited.

2. The main theoretical result extends an analogous result by Kennedy (2020), which somehow doesn’t get mentioned until Section 4.2. Moreover, citations for ITE without unobserved confounding could be more generous; it is a vast literature.

3. To distinguish the originality of this paper relative to Syrgkanis et al (2019), I would like to see a discussion that compares the estimation procedures and their guarantees. If I understand correctly, the Syrgkanis et al (2019) estimator proposes a Neyman orthogonal loss while this work used the pseudo-outcome technique, yet both provide excess risk rates that involve products of nuisance rates.

4. Estimation that uses the efficient influence function to study ITE in the IV setting has been proposed by Ogburn et al (2015). Estimation that combines the efficient influence function with sample splitting and machine learning has been analyzed in the semiparametric IV setting, e.g. Singh and Sun (2019). This paper goes further than these previous works in proposing an ML estimator for ITE. I would like these previous works to be cited as well.

Quality

1. I will raise the score if the experiments are made to align more closely with the theory.

-line 284. Is there any sense in which the proposed nuisance estimators would satisfy Assumption 3?

-line 294. I would like to see the experiments with sample splitting since that is the proposal in the theory section

-line 313. The comparisons are hard to understand. The various estimators have different estimands. For example, DeepIV estimates a function that is not the ITE. So what is being reported here? These comparison estimators also involve sample splitting in some cases, to align with their theory, while the proposed estimator is not implemented with sample splitting (though its theory requires it)

2. I would like more discussion in the case study. In line 386, at what values are these variables fixed? Simply fixing values means that the quantity visualized in Figure 1 is not the ITE. Nor is it the ITE conditional on some subset of covariates. Please interpret what is being displayed.

3. The theoretical result, stated in Section 4.2 and explained in Section 4.3, is well written and interesting. I would like to see the additional assumptions of Theorem 2 stated in the main text, rather than pointing to another paper. Then the reader can assess whether they are “mild” or not.

Clarity

1. There were several instances in which the authors made statements that were either unclear or untrue as written. I will increase the score when these are addressed.

-line 25. The definition of unconfoundedness is incorrect. The given definition is an example of when unconfoundedness holds.
-line 92. Exclusion is not unrestrictive.

-line 107. The descriptions in Assumption 2 are misleading. The assumptions state that U can essentially be differenced out. It is an additive equi-confounding.

-line 117 (footnote). The definition of compliers is incorrect. It is the subpopulation for whom A(1)>A(0).

-line 142. This statement is misleading. Arguably this paper requires a type of additive separability of confounding as well; see the comment above.

-line 223. This is a misleading citation. Cross-fitting is a type of sample splitting that has the same theoretical analysis. It is not a practical shortcut. The cited paper does not advocate fitting all nuisances on the same data set; that approach is only valid when the function spaces have limited complexity.

2. Please provide some intuition for the first term in equation 3. It helps to group \hat{\tau}_{init} and look at what residual is being extracted from Y before being reweighted.

Significance

My assessment of significance will hinge upon the authors’ improvements described above. There are currently too many issues for this paper to be a significant contribution, but I am optimistic that these issues can be addressed in the revision.

---

> ### Author Response · Authors · 2022-08-02
> **Responses to Reviewer biU1**
>
> Thank you for your review and your helpful comments! We improved our work in the following ways:
> 1. We improved the framing of our contribution by discussing the differences between our framework and previous work more clearly.
> 2. We also included additional citations as per your suggestion. These have helped us in showing the existing research gap and how our paper is novel.
> 3. We followed your suggestion and aligned the experiments with our theory by adding new results for DRIV and our MRIV using cross-fitting. Our new results again establish that our MRIV framework is superior (see new Appendix I).
> We highlighted these (and other changes) in red color in our revised paper (see rebuttal PDF).
>
> ### Responses to "Originality"
>
> 1. We followed your feedback and cited the suggested references. For this, we have added a new paragraph to our Related Work section on existing doubly robust estimators (see Section 3, "Doubly robust IV methods"). Therein, we have now included the citations of Kennedy et al (2019) and Semenova and Chernozhukov (2021). We also emphasize that these papers propose **doubly** robust methods for different settings and/or different estimands. In contrast, our MRIV is the first **multiply** robust method for ITE estimation in the **binary** IV setting (see our new Table 1).
>
> 2. We followed your suggestion closely and included additional citations to Kennedy (2020) in both our Introduction and Related Work sections (see Sections 1 and 3). However, to the best of our knowledge, we are the first to adapt their approach to the *IV setting*. We also added additional citations of methods for other unobserved confounding settings to our literature section.
>
> 3. Differences to DRIV (Syrgkanis et al 2019) and how our MRIV is novel: We improved our paper by clarifying the differences of our framework and DRIV (Syrgkanis et al 2019) in both the Related Work section and Appendix E. There are two key differences between the papers: (i) Our framework is **multiply robust**, while DRIV is only **doubly robust**. (ii) We derive a **multiple robust** convergence rate, while the rate in Syrgkanis et al 2019 is not robust with respect to the nuisance rates. We clarify the differences throughout our revised paper (see our changes in the Introduction, Related Work section, and Appendix E in the rebuttal PDF).
>
> 4. We followed your suggestion and cited Ogburn et al (2015) and Singh and Sun (2019)) in the Related Work section. Similar to Syrgkanis et al 2019, both methods are doubly robust but do **not** achieve multiple robustness properties.
>
> ### Responses to "Quality"
>
> 1. **Better alignment between theory and experiments:** We found your feedback very helpful and invested additional effort to align our experiments more closely with the experiments (see new Appendix I).
>
> - *Assumption 3* We stated the smoothness assumption mainly to illustrate how the rate of MRIV depends on the nuisance rates. Notably, we are not dependent on smoothness but derived a general bound of the asymptotic MSE depending on the MSE of the nuisance parameters (see the proof of Theorem 2). Hence, using different assumptions on the nuisance rates would lead to different bounds. We have provided a theoretical example under sparsity assumptions in Appendix B.
>
> - *New results for cross-fitting:* We performed **additional experiments** with cross-fitting (for both MRIV and the respective baselines) to make the experiments section more in line with the theory. We added our results to Appendix I of our revised paper. Importantly, the results confirm the effectiveness of our proposed method.
>
> - *Baseline comparison:* It is true that not all baselines are designed for ITE estimation but rather for predicting counterfactual outcomes. We included them nevertheless to compare a comprehensive set of baselines. For these baselines, we worked with the differences between the predicted factual and the counterfactual outcomes. We added a clarification to Section 5 of the main text and to the implementation details in Appendix E.
>
> 2. **Case study:** We greatly appreciate the opportunity to expand our discussion of the case study. In Figure 5, we visualize the treatment effect heterogeneity with respect to age and gender, on the English-speaking subpopulation that signed up alone with the number of emergency visits fixed to one. More formally, we first estimate the ITE $\hat{\tau}(\mathrm{Age}, V)$, where $V$ denotes the other covariates. Then, we plot the function $\hat{\tau}(\cdot, V = v)$. The plot shows that the results from our method are consistent with other IV methods in the sense that it estimates larger causal effects for older ages. To improve our paper, we added this discussion to our case study (see new materials in Section 5).
>
> 3. We added the additional assumption from Kennedy (2020) to Theorem 2 of our revised paper.
>
> ### Responses to "Clarity"
> We thank you for the suggestions to improve clarity and followed all of them.

---

> > ### Comment · Reviewer_biU1 · 2022-08-09
> > **Thank you for the response**
> >
> > The authors' updated draft addresses all of the points that I raised. I will raise the score accordingly

---

### Official Review · Reviewer_2sYo · 2022-07-16

**Rating:** 7
**Confidence:** 3
**Soundness:** 3 good
**Presentation:** 3 good
**Contribution:** 3 good

**Summary:**

This paper studies a multiply-robust estimate of the conditional average treatment effect (CATE), $ \tau(x) = \mathbf{E}[Y(1) - Y(0) \mid X=x] $ using a binary instrumental variable (IV) for identification. The approach is to construct a transformed outcome, $\hat{Y}_0$ such that $E[ \hat{Y}_0 \mid X=x] = \tau(x)$, and then use a machine learning regression model to estimate the mean of $\hat{Y}_0$ using $X$. The paper shows that such an approach can improve convergence rates over naive approaches such as plug-in estimates typically used for IVs.

**Questions:**

How is this work different from DRIV method of Syrgkanis et al., 2019?

**Strengths And Weaknesses:**

The paper is reasonably well written. While it focuses a lot of attention on the convergence rates, which require following a lot of subtlety that makes it difficult for the reader to fully understand the main message, it is reasonably high quality and appropriate for publication on the topics of clarity and quality.

Unfortunately, the authors seem to have missed a key paper on this topic. Their abstract says "[t]o the best of our knowledge, our MRIV is the first multiple robust machine learning framework tailored to estimating ITEs in the binary IV setting." This is strange, as they cite  Syrgkanis et al., 2019 [29 in the paper], who provide almost an identical method for inference (up to some algebra and mild re-parameterization), show that it is multiply robust, and provide convergence rates under certain conditions, similarly emphasizing the dependence on the nuisance parameter convergence rates.

Edit: it appears that "multiple" (meaning more than double) robustness is the key innovation here, and that there is a nontrivial practical difference between it and double robustness. See the conversation in the reviews for more details. While the transformed outcomes are very similar, and have the same conditional mean, suggesting some way of transforming from one to the next, the differences are nontrivial, and allow for construction of different nuisance parameter estimates that allow for improved performance. The authors have clarified this in their updated version, and therefore, I believe the paper is appropriate for publication.

---

> ### Author Response · Authors · 2022-08-02
> **Response to Reviewer 2sYo**
>
> Thank you for your review! We welcome the opportunity to clarify the difference between DRIV (Syrgkanis et al (2019)) and MRIV (our framework). In the following, we provide a detailed comparison between the two methods and highlight our contribution. We also added this comparison to the revised version of our paper (see new changes to Section 1, Section 3, and Appendix E) and highlighted these changes (and other) in red color (see rebuttal PDF).
>
> There are two key differences between our paper and Syrgkanis et al (2019): (i) Our framework (MRIV) is **multiply robust**, while DRIV is only **doubly robust**. (ii) We derive a **multiple robust convergence rate**, while the rate in Syrgkanis et al (2019) is not robust with respect to the nuisance rates.
>
> Ad (i): Both MRIV and DRIV perform a pseudo-outcome regression on the efficient influence function (EIF) of the ATE. The key difference: DRIV uses the *doubly robust parametrization* of the EIF from Okui et. al. (2012),
> whereas we use the *multiply robust parametrization* from Wang and Tchetgen Tchetgen (2018). Hence, our MRIV frameworks extends DRIV in a non-trivial way to achieve multiple robustness (rather than double robustness).
> Hence, our estimator is consistent in the union of three different model specifications. We also refer to Wang and Tchetgen Tchetgen (2018), Section 4.5, for a detailed discussion on multiple robustness and the importance of the EIF parametrization. We clarify the difference between multiply and doubly robust methods throughout our revised paper (see our changes in the Introduction, Related Work section + Table 1, and Appendix E in the rebuttal PDF).
>
> Ad (ii): It is true that Syrgkanis et al (2019) provide a similar convergence rate that depends on the nuisance parameter convergence rates using Neyman orthogonality. However, their rate is
> **not** robust in the sense that their estimator does **not** achieve a fast rate even when some nuisance parameters converge slowly. We detail this property in the following analysis. We also add the corresponding analysis to Appendix E of our paper.
> Let us assume that the pseudo regression function is $\gamma$-smooth and that we use the same second-stage estimator $\hat{\mathrm{E}}_n$ with minimax rate $n^{-\frac{2\gamma}{2\gamma + p}}$ for both DRIV and MRIV.
>
> DRIV takes the initial ITE estimator $\hat{\tau}_{\mathrm{init}}(x)$ as input and estimates the nuisance parameters $q(X) = \mathrm{E}[Y \mid X]$, $p(X) = \mathrm{E}[A \mid X]$, $f(X) = \mathrm{E}[A \cdot Z \mid X = x]$, and $\pi(X) = \mathrm{E}[Z \mid X = x]$. Assume that all nuisance parameters are $\alpha$-smooth, and are estimated with minimax rate $n^{\frac{-2\alpha}{2\alpha+p}}$. Then, Corollary 4 from Syrgkanis et al (2019) states the DRIV rate
> \begin{equation}
>    n^{\frac{-2\gamma}{2\gamma+p}} + n^{\frac{-4\alpha}{2\alpha+p}}.
> \end{equation}
>
> For our Theorem 2, we assume estimation of the nuisance parameters $\mu_0^Y(x)$ with rate $n^{\frac{-2\alpha}{2\alpha+p}}$, $\mu_0^A(x)$ and $\delta_A(x)$ with rate $n^{\frac{-2\beta}{2\beta+p}}$, and $\pi(x)$ with rate $n^{\frac{-2\delta}{2\delta+p}}$. If $\hat{\tau_{\mathrm{init}}}(x)$ converges with rate $r_{\tau}(n)$, our Theorem 2 yields the rate
> \begin{equation}
>    n^{\frac{-2\gamma}{2\gamma+p}} + r_{\tau}(n) \left(n^{\frac{-2\beta}{2\beta+p}} + n^{\frac{-2\delta}{2\delta+p}}  \right) +
>     n^{-2\left(\frac{\alpha}{2\alpha+p}+\frac{\delta}{2\delta+p}\right)} +
>     n^{-2\left(\frac{\beta}{2\beta+p}+\frac{\delta}{2\delta+p}\right)} .
> \end{equation}
> If all nuisance parameters converge with the same rate of $n^{\frac{-2\alpha}{2\alpha+p}}$, the rates of DRIV and MRIV coincide. However, different to DRIV, our rate is multiply robust in the spirit of Theorem 1. For example, if $\delta$ is small, our rate is still fast as long as $\alpha$ and $\beta$ are large (i.e., if the other nuisance parameters are sufficiently smooth).
>
> As a toy example of when our rate is faster, consider the case where $\mu_0^Y(x) = \mathrm{E}[Y \mid X, Z = 0]$ is very smooth, but $\mu_1^Y(x) = \mathrm{E}[Y \mid X, Z = 1]$ is not. In this case, the nuisance parameter $q(X) = \mathrm{E}[Y \mid X]$ of DRIV will be hard to estimate, resulting in a slow rate of convergence. In contrast, our MRIV rate will be fast as long as $\beta$ and $\delta$ are large or $\beta$ is large and the initial estimator converges fast.
> Also, note that we do not necessarily need to make smoothness assumptions: Our rate builds on a general bound of the MSE which we derive in Appendix A.
>
> To improve our paper, we made substantial changes to our Introduction, the Related Work section, and our new Conclusion. We also added a new Table 1 with a literature comparison to highlight our novelty:
> our framework is the *first for **multiply robust*** ITE estimation. For technical details, we also added an in-depth comparison to our new Appendix E.

---

### Author Response · Authors · 2022-08-02
**Response to all reviewers**

Thank you very much for you reviews! We closely followed your suggestions uploaded a revised versions of our paper and our appendix (see rebuttal PDF). We highlighted our changes in red color. We addressed all of your comments, and find that our paper has improved substantially as a result.

### Summary of our contributions

Here, we briefly repeat the main contributions of our work:
1. We propose a novel, multiply robust machine learning framework (called MRIV) to learn the ITE in the binary IV setting. To the best of our knowledge, ours is the first that is **multiply robust**. As a result, our MRIV is consistent in the union of three model specifications. For comparison, existing works for ITE estimation (such as DRIV from Syrgkanis et al (2019)) are only **doubly robust**.
2. We prove that MRIV achieves a **multiply robust convergence rate**. This is in contrast to the rate in Syrgkanis et al (2019), which is not robust with respect to the nuisance rates. We further show that our MRIV is asymptotically superior to existing plug-in estimators. 3. We propose a tailored deep neural network, called MRIV-Net, which builds upon our framework to estimate ITEs. We demonstrate empirically that MRIV-Net achieves state-of-the-art performance.

### Key improvements

We followed all suggestions of the reviewer team and improved our paper in the following ways:
1. **Novelty of our method**: We state more clearly why our method is different from Syrgkanis et al (2019) and why it is thus novel. Importantly, the DRIV method by Syrgkanis et al (2019) is only **doubly** robust. In contrast to that, our MRIV method is the \underline{first} that is **multiply** robust. To clarify this, we added a new Table 1 highlighting our novelty. Our framework is a non-trivial contribution and requires a completely different methodological approach than DRIV (e.g., different algorithm, different theory, different derivations, etc.). Reassuringly, we remind that we included DRIV as a baseline. In our results, we show that we outperform DRIV by **a large margin**.
-> To improve our paper, we clarified the difference between their work and ours throughout our paper (see our new Table 1). Specifically, we made substantial changes to our Introduction (see new materials in Section 1), Related Work (see new materials in Section 3), and Conclusion (see new Section 6) to point out salient differences. We further added a new comparison to Appendix E where we provide a detailed, technical comparison of how and why the convergence rates are different.

2. **More background on doubly vs. multiply robust methods** We provided an in-depth explanation on the differences of doubly vs. multiply robust methods. Importantly, existing work for ITE estimation using instruments is only *doubly robust*. In contrast, ours is the first *multiply robust* estimator. Accordingly, we greatly expand our Related Work (see new materials in Section 3) and clarify key differences. For better readability, we also added a new Table 1 summarizing the differences between doubly vs. multiply robust methods and how our method is novel.
3. **New results:** We performed new experiments with cross-fitting. This allowed us to align our experiments more closely to our theory. We added our results as a new Appendix I. Importantly, our results again confirm the effectiveness of our proposed method.

We also addressed all other suggestions and, as a result, made various clarifications throughout our paper. We detail the changes in the corresponding point-by-point responses.

In sum, we took all of your suggestions at heart and made substantial revisions. We are confident that we remedied all weaknesses as a result and are convinced that our paper is a valuable contribution to the literature. We hope you agree.

---

> ### Comment · Reviewer_2sYo · 2022-08-09
> **Multiple robustness**
>
> Dear authors,
>
> Thanks for your responses and updates to the paper. I think it is more clear now. I do have some issues with the emphasis on multiple robustness, and questions about it's significance. However, it does seem that there are some practical advantages to this formulation, and the contextualization is more clear now, so I will increase my score.
>
> In general regarding multiple robustness, I wonder if it's really clear that it is an improvement over double robustness. Whereas double robustness requires only 1 of 2 models to be correctly specified, multiple robustness (in this paper) requires 2 of 3 to be correctly specified. Related to this, it seems like there might be some overspecification of the nuisance parameters, and therefore some constraints on the smoothness parameters? For example, does having a specific smoothness of $\mu^Y_a(\cdot)$ imply something about the best possible initial rates for $\hat\tau_{\text{init}}(\cdot)$?
>
> From a readability perspective, it might be helpful to have a table for all of the nuisance parameters, their names, their definition as a conditional expectation, their associated smoothness parameter, and how the smoothness is related to the smoothness of other parameters. This would make it easier for the reader to parse the theorems without scrolling around to look for which smoothness parameter is associated with which nuisance parameter, and would help clarify the above point.

---

### Meta-Review · Area_Chair_mDtA · 2022-08-27

**Recommendation:** Reject
**Confidence:** Certain

**Metareview:**

The authors offer a methodologically novel and very interesting approach to an important problem of CATE (or conditional LATE) with binary instrument/treatment. The paper should be published at a good ML venue.

Unfortunately, I need to recommend rejection primarily because the key proofs supporting their main theorems were missing from the original supplementary material and were only submitted after the rebuttal. Given that these were crucial elements supporting their main results and that they were submitted post the original submission deadline it seems unfair (even if it was most probably an honest mistake of the authors).

Another issue that came up in the discussion phase that seems crucial to revise before a resubmission is that the authors main estimation rate result relies on a main theorem of the unpublished work of Kennedy 2020, which has since been revised and the new theorem in Kennedy 2020, is technically very different and requires different assumptions. Hence the authors need to revise their main estimation rate result accordingly.

However, I acknowledge that this is not the main contribution of this work and a simple invocation of past work, but rather the main contribution is to formulate a loss for CATE using the idea of Want and Tchetgen-Tchetgen. So this wouldn't most probably be a reason for rejection.

**Award:**

No

---

### Decision · Program_Chairs · 2022-09-14

Reject